# Differential effects of 40S ribosome recycling factors on reinitiation at regulatory uORFs in *GCN4* mRNA are not dictated by their roles in bulk 40S recycling
Kristína Jendruchová [1,2,4], Swati Gaikwad [3,4], Kristýna Poncová[1], Stanislava Gunišová[1], Leoš Shivaya Valášek [1] ✉ & Alan G. Hinnebusch [3] ✉

Recycling of 40S ribosomal subunits following translation termination, entailing release of deacylated tRNA and dissociation of the empty 40S from mRNA, involves yeast Tma20/Tma22 heterodimer and Tma64, counterparts of mammalian MCTS1/DENR and eIF2D. MCTS1/DENR enhance reinitiation (REI) at short upstream open reading frames (uORFs) harboring penultimate codons that confer heightened dependence on these factors in bulk 40S recycling. Tma factors, by contrast, inhibited REI at particular uORFs in extracts; however, their roles at regulatory uORFs in vivo were unknown. We examined effects of eliminating Tma proteins on REI at regulatory uORFs mediating translational control of *GCN4* optimized for either promoting (uORF1) or preventing (uORF4) REI. We found that the Tma proteins generally impede REI at native uORF4 and its variants equipped with various penultimate codons regardless of their Tma-dependence in bulk recycling. The Tma factors have no effect on REI at native uORF1 and equipping it with Tma-hyperdependent penultimate codons generally did not confer Tma-dependent REI; nor did converting the uORFs to AUG-stop elements. Thus, effects of the Tma proteins vary depending on the REI potential of the uORF and penultimate codon, but unlike in mammals, are not principally dictated by the Tma-dependence of the codon in bulk 40S recycling.

Synthesis of proteins is carried out by ribosomes, molecular machines that assemble from the small (40S) and large (60S) ribosomal subunits. During translation initiation, the 40S subunit acquires the ternary complex (TC) composed of the methionyl initiator Met-tRNA$_i^{Met}$, GTP and the eukaryotic Initiation Factor (eIF) 2, forming the 43S pre-initiation complex (PIC). Upon mRNA recruitment, the 48S PIC thus formed scans the mRNA 5' leader region to locate the authentic initiation codon, generally an AUG triplet in a favorable "Kozak" sequence context. Subsequently, subunit joining occurs to produce the 80S initiation complex and elongation commences. Upon stop codon recognition, translation terminates, and the nascent peptide is released. This is followed by ribosomal recycling, i.e.,

splitting of both ribosomal subunits and dissociation of the 40S and the deacylated tRNA that decoded the last sense codon from mRNA, to start the new translational cycle (reviewed in refs. [1]–[4]). In some cases, the recycling step is incomplete, leaving the post-termination 40S mRNA-bound, which allows it to resume traversing downstream and, upon reacquisition of the TC, reinitiate translation of a downstream ORF (reviewed in ref. [5]).

Translation termination and ribosome recycling are intrinsically linked. When the stop codon enters the ribosomal A site, it is recognized by a heterodimer composed of eukaryotic Release Factors 1 and 3 (eRF1/eRF3). Stop codon recognition results in GTP hydrolysis on eRF3 followed by its release. eRF1 then catalyzes hydrolysis of the nascent polypeptide chain,

[1]Laboratory of Regulation of Gene Expression, Institute of Microbiology of the Czech Academy of Sciences, Videnska 1083, 142 20 Prague, Czech Republic.
[2]Faculty of Science, Charles University, Albertov 6, 128 00 Prague, Czech Republic. [3]Division of Molecular and Cellular Biology, Eunice Kennedy Shriver National Institute of Child Health and Human Development, National Institutes of Health, Bethesda, MD, 20892, USA. [4]These authors contributed equally: Kristína Jendruchová, Swati Gaikwad. ✉e-mail: valasekl@biomed.cas.cz; alanh@mail.nih.gov

which generates the post-termination 80S ribosome harboring deacylated tRNA base-paired with the penultimate codon in the P site[6]. The ATP-binding cassette protein Rli1/ABCE1 (yeast/mammals) facilitates the dissociation of the 60S subunit both in vitro in yeast and mammalian reconstituted systems[7,8] and in vivo in yeast cells[9], aided by the eIF3-associated factor eIF3j[10,11]. Studies in vitro revealed that after subunit splitting, the deacylated tRNA and mRNA are dissociated from the 40S post-termination complex by the non-canonical initiation factor eIF2D or its functionally and structurally related heterodimer MCTS1/DENR[12,13]. DENR and MCTS1 bear sequence similarity to the N-terminal and C-terminal regions of eIF2D, respectively. The MCTS1 and the eIF2D N-terminal region harbor DUF1974 and PUA domains, whereas DENR and eIF2D's C-terminal region contain SWIB/MDM2 and SUI/eIF1 domains. eIF2D additionally contains a central winged-helix domain[4].

Ribosome profiling studies demonstrated that the yeast orthologs of eIF2D, MCTS1 and DENR, known as Tma64, Tma20 and Tma22, respectively[14], function in 40S post-termination complex recycling in vivo[15,16]. Thus, deleting the corresponding genes led to accumulation of unrecycled 40S subunits at the majority of stop codons, with the largest defect observed in Δ*tma20*Δ*tma64* (*tma*ΔΔ) cells. Quantifying the accumulation of unrecycled 40S subunits at all individual stop codons in *tma*ΔΔ vs. wild-type (WT) cells revealed that certain penultimate codons result in significantly higher accumulation of unrecycled 40S subunits in the mutant cells—a phenomenon dubbed Tma-dependence. Furthermore, comparison of the double deletant to single deletant strains showed that Tma64 is largely dispensable, whereas both subunits of the Tma20/Tma22 heterodimer are required for the majority of 40S recycling events in vivo. A notable consequence of defective 40S recycling in mutants lacking an intact Tma20/Tma22 heterodimer is increased REI downstream of stop codons of main ORFs within the 3' untranslated regions (3' UTRs) of mRNAs. Such non-canonical REI frequently occurs by conventional scanning of the unrecycled post-termination 40S complexes to 3' UTR-situated AUG codons, preferentially in optimum Kozak context[15,16]. Absence of the Tma20/Tma22 heterodimer also conferred increased canonical REI downstream of the stop codons of upstream open reading frames (uORFs) in the 5' UTRs of reporter mRNAs in yeast cell-free translation extracts[15].

Whereas yeast Tma20/Tma22 were shown to inhibit REI, work on their mammalian orthologs indicated that human DENR conversely promotes REI after translation of certain uORFs, including 1aa-long uORFs comprised of only a start and stop codon, referred to here as "start-stops", with their ATG start codons in optimum Kozak context[17,18]. In the absence of DENR/MCTS1, mRNAs containing start-stops exhibit reduced translation of the downstream main ORF. DENR-dependence for REI was also established for certain other short uORFs in a manner generally dictated by the presence of particular penultimate codons that, in 40S ribosome profiling data, were found to confer heightened dependence on DENR for bulk 40S recycling at stop codons throughout the translatome[13]. Indeed, the degree of DENR-dependence often inversely correlates with the propensity of the deacylated tRNA decoding the penultimate codon to dissociate spontaneously from post-termination 40S complexes in reconstituted termination/recycling systems[13,19]. Accordingly, it was proposed that only those uORFs harboring penultimate codons with low rates of spontaneous dissociation of the cognate tRNAs require DENR/MCTS1 to accelerate release of the deacylated tRNA and allow the post-termination 40S to resume traversing downstream and reacquire the TC at the now-empty P site, enabling REI. The DENR-requirement for REI included the regulatory short uORF that stimulates REI on *ATF4* mRNA (uORF1), allowing scanning ribosomes to bypass a more distally positioned uORF (uORF2) that inhibits initiation at the main CDS, both in mammalian cells[13] and fruit flies[20].

To reconcile the apparent discrepancy between yeast Tma factors and their mammalian counterparts, it was suggested[13] that release of the deacylated tRNA by MCTS1/DENR does not lead to dissociation of post-termination 40S subunits at relatively short uORFs in mammalian cells owing to retention of eIF3, eIF4G1, or eIF4E by 80S ribosomes translating

the uORFs and continued occupancy of these factors on the post-termination 40S subunits[21-23]. These retained eIFs presumably impede the 40S dissociation function of MCTS1/DENR, making the only observable consequence of depleting DENR a reduction in REI at the subset of uORFs requiring DENR for efficient release of the deacylated tRNA, whose continued presence in the P site in the absence of DENR then prevents the 40S post-termination complexes from traversing downstream or recruiting TC to the still-occupied P site. Depleting DENR has no effect on REI at the short uORFs where deacylated tRNA dissociates independently of MCTS1/DENR. These alternative outcomes are depicted schematically in Fig. 1A(i)-(ii). At typical uORFs in yeast, by contrast, release of the deacylated tRNA, whether or not it is highly dependent on the Tma proteins, would lead to subsequent dissociation of post-termination 40S subunits from mRNA and low-level REI following most uORFs in WT cells, because eIFs are generally not retained by 80S ribosomes translating yeast uORFs[24-27]. As such, a further reduction in REI in cells lacking Tma factors is not expected at typical yeast uORFs even when they harbor penultimate codons highly dependent on the Tma factors to stimulate release of deacylated tRNAs from the post-termination 40S subunits. These expected outcomes for typical uORFs in yeast are shown in Fig. 1B(i)-(ii).

A different outcome would be predicted for a handful of atypical uORFs in yeast, where 40S dissociation is counteracted by initiation factors that remain associated with post-termination 40S complexes. The 5'-proximal AUG-initiated uORF in *GCN4* mRNA, uORF1, is the best characterized such uORF in yeast. It is optimized for eIF3 and eIF4G binding and retention of post-termination 40S subunits by the presence of cis-acting REI promoting elements (RPEs) rendering it highly permissive for REI downstream (Fig. 2)[5,27,28]. Accordingly, the 40S post-termination complexes at the uORF1 stop codon should not spontaneously dissociate from mRNA on release of the deacylated tRNA, as described above for short uORFs in mammalian cells (Fig. 1A). This in turn should enable us to determine whether eliminating the Tma proteins confers reduced REI at uORF1 variants equipped with penultimate triplets with heightened dependence on Tma factors for release of deacylated tRNA in bulk 40S recycling. At the same time, we should observe little or no effect at uORF1 variants containing penultimate codons that are less dependent on the Tma proteins in bulk 40S recycling, in the manner depicted in Fig. 1A(i)-(ii) for DENR-depletion in mammalian cells.

Importantly, *GCN4* uORF1 is functionally equivalent to uORF1 in *ATF4* mRNA, and *GCN4* and *ATF4* translation are governed by very similar REI mechanisms[5]. However, whereas *ATF4* uORF1 is DENR-dependent for efficient REI[13,20], we determined previously that translational control of WT *GCN4* mRNA occurs normally in the *tma20*Δ*tma64*Δ mutant[29]. This finding implies that the Tma20/Tma22 heterodimer is not required to dissociate deacylated tRNA$^{Cys}$ from the native penultimate UGC codon of WT *GCN4* uORF1. Consistently, UGC was judged to be hypodependent on DENR in mammalian cells[13] and one of the least Tma-dependent codons for bulk 40S recycling in yeast[16]. There is also evidence that the cognate tRNA$^{Cys}$ is weakly associated with mammalian 40S post-termination complexes[19]. Here, using uORF1 as a genetic tool, we examined whether substituting the native UGC codon in *GCN4* uORF1 with different codons judged to be Tma-hyperdependent for bulk 40S recycling will reduce REI in *tma*ΔΔ cells in the manner observed for DENR-dependent short uORFs in DENR-depleted mammalian cells (Fig. 1A(i)-(ii)).

In contrast to REI-permissive uORF1, the fourth AUG-initiated uORF in *GCN4* mRNA (uORF4) does not allow retention of eIFs on post-termination 40S subunits, owing to multiple, surrounding cis-acting sequences, thus making it non-permissive for REI and repressive to *GCN4* translation in WT cells[26,27,30–33]. We reasoned that uORF4 is a typical example of short uORFs in yeast that do not allow REI owing to efficient dissociation of 40S post-termination complexes regardless of the requirement for Tma factors in releasing tRNA (Fig. 1B(i)-ii)). uORF4 contains a penultimate CCG proline codon, whose dependence on DENR for recycling and REI in human cells is ambiguous[13], and was neither unusually dependent nor independent of the Tma proteins for

**Fig. 1 | Predicted mechanisms governing MCTS1/ DENR mediated recycling and REI at mammalian short uORFs and expected functions of yeast Tma20/Tma22 at typical REI-non-permissive yeast uORFs.** MCTS1/DENR or Tma20/Tma22 are depicted as linked blue ovals whose distinct functions in releasing the penultimate deacylated tRNA (red "L") and subsequently dissociating the empty 40S from the mRNA are labeled with boxes 1 and 2, respectively, and solid or dashed arrows. **A** DENR-mediated recycling at short mammalian uORFs depending on the presence of (i) DENR-hyperdependent penultimate codons requiring the recycling factors for efficient tRNA release, or (ii) DENR-hypodependent codons where tRNA can be released at high levels spontaneously. (i) DENR-hyperdependent uORFs: in WT cells, "(+) DENR", MCTS1/DENR efficiently releases the penultimate-codon tRNA (function 1) but its second function in 40S dissociation is impeded by eIF3 (green oval) retained on post-termination 40S subunits to enable high-level REI at the main CDS. In cells depleted of DENR, "(−) DENR", diminished release of the penultimate-codon tRNA lowers REI. (ii) DENR-hypodependent uORFs: efficient release of the tRNA and high-level REI occur both in the presence (upper) or absence (lower) of DENR, yielding no decrease in REI on DENR depletion. **B** Tma-mediated recycling at typical yeast short uORFs depending on the presence of (i) Tma-hyperdependent or (ii) Tma-hypodependent penultimate codons. (i) Tma-hyperdependent uORFs: in WT cells (upper), Tma factors efficiently release the penultimate-codon tRNA (function 1) but also dissociate the empty 40S subunit from mRNA (function 2), conferring low-level REI. In *tma∆∆* cells (lower), the absence of Tma-mediated tRNA release helps to ensure low-level REI, yielding no change in REI in *tma∆∆* vs. WT cells. (ii) Tma-hypodependent uORFs: Tma-independent tRNA release occurs efficiently in both WT and *tma∆∆* cells, but REI is low in both cases because, lacking eIF3, the empty 40S subunits dissociate from the mRNA independently of Tma factors, for no change in REI in *tma∆∆* vs. WT cells.

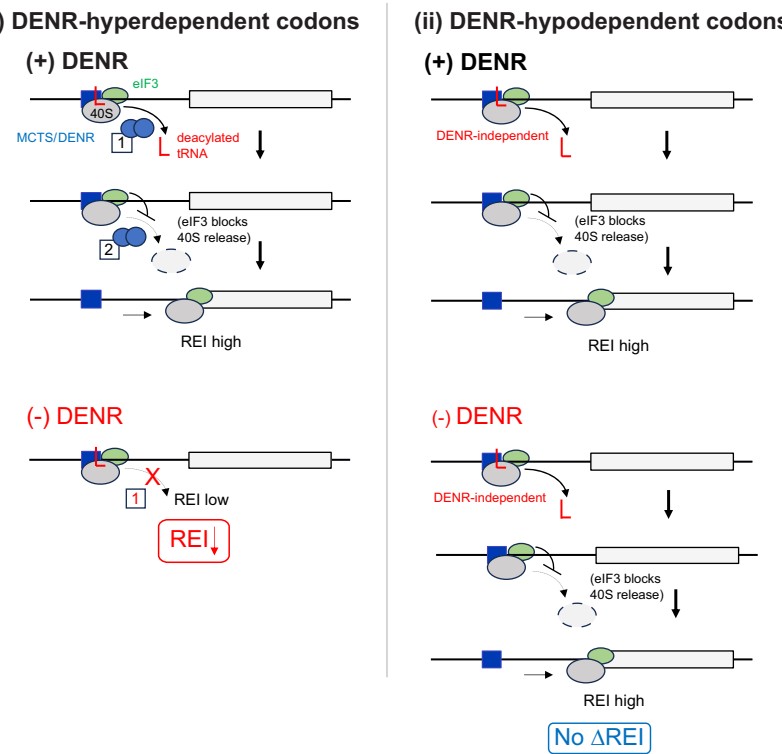

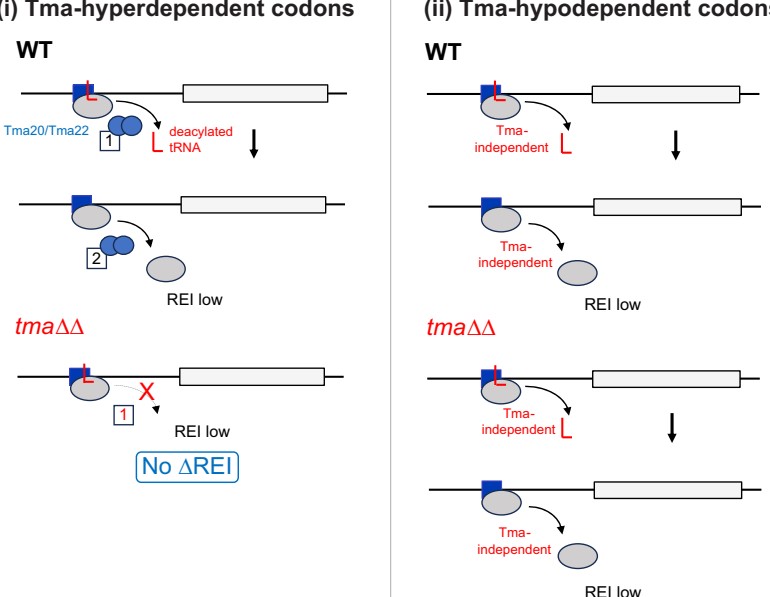

bulk 40S recycling in yeast[16]. Since elimination of the Tma20/Tma22 heterodimer did not impact translational control of WT *GCN4* mRNA[29], as mentioned above, it seems unlikely that CCG contributes to low-level REI at WT uORF4 by imposing a heightened requirement for Tma factors for tRNA release. Here, using uORF4 as another genetic tool, we tested the expectation that replacing the native CCG codon with other codons known to be hyperdependent on Tma proteins for bulk recycling

will not reduce REI in *tma∆∆* cells because, lacking eIF3, the 40S subunits will dissociate and fail to reinitiate regardless of the tRNA release kinetics (Fig. 1B(i), (ii)). We also examined whether converting the native 3-codon uORF1 and uORF4 to start-stop elements by eliminating their second and third coding triplets confers a dependence on the Tma proteins for REI in the manner observed for DENR with these specialized uORFs in mammalian cells[17,18].

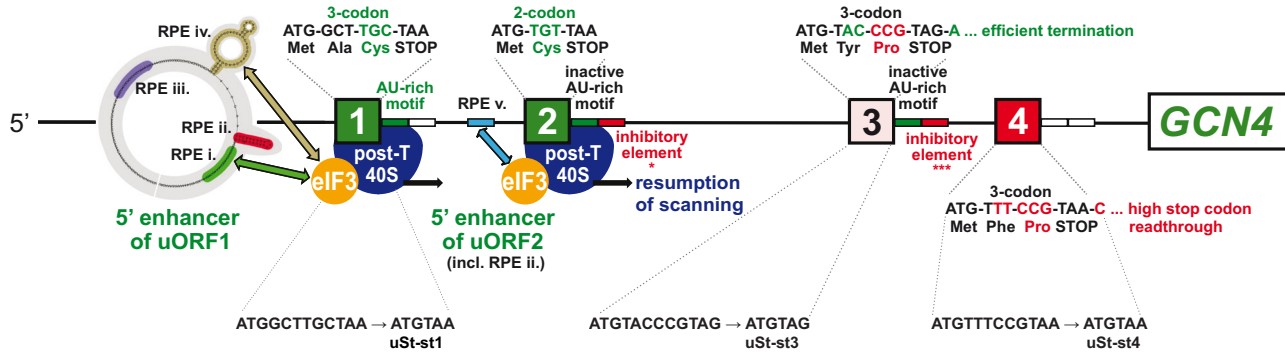

**Fig. 2 | Summary of all *cis*-determinants that either promote or inhibit REI on *GCN4* mRNA after translation of its four short uORFs.** Schematics of the 5′ enhancers of uORF1 and 2 containing their respective RPEs, some of which functionally interact with eIF3 to promote resumption of scanning. Green color-coding generally indicates stimulatory effects of the corresponding *cis*-factors on efficiency of REI, whereas red color-coding indicates inhibitory effects (with the exception of RPE ii. of uORF1, which is also stimulatory); the number of asterisks below the inhibitory elements of the uORF2 and uORF3 3′ sequences depicts the degree of their inhibition as determined experimentally. Mutations converting uORFs 1, 3, or 4 to Start-stop elements uSt-st1, uSt-st3, or uSt-st4, described later in RESULTS, are given below the respective WT uORF schematics. Reprinted and modified with permission from ref. 33.

Our results depart from the expected outcomes for *GCN4* uORF4 shown in Fig. 1B and suggest instead a role for Tma factors at this uORF in dissociating 40S post-termination complexes from mRNA following release of deacylated tRNA, regardless of whether the penultimate triplets are Tma-hyperdependent in bulk recycling. Our findings for *GCN4* uORF1 also differ from those obtained for short, REI-permissive uORFs in mammalian cells[13] (Fig. 1A(i), (ii)) in revealing that uORF1 generally allows efficient REI independently of Tma factors even when equipped with Tma-hyperdependent penultimate codons. Overall, the effects of Tma factors on REI at the functionally distinct *GCN4* regulatory uORFs equipped with different penultimate codons are not principally dictated by the Tma-dependence of these codons in bulk recycling. We also found that, unlike in mammalian cells, REI following start-stop uORFs is Tma-independent; and we further show that Tma64 plays a supportive role to that of Tma20/Tma22 in impeding REI at uORF4, consistent with its auxiliary role in bulk 40S recycling at most stop codons in yeast.

## Results

### Tma64 partially substitutes for the Tma20/Tma22 heterodimer in suppressing REI after translation of the *GCN4* uORF4 variant containing the Tma-hyperdependent TTG^Leu codon

To examine the effects of replacing the penultimate codons of uORF1 or uORF4 with triplets that differ in their dependence on Tma factors in bulk 40S recycling, we employed the well-established *GCN4-lacZ* reporter system, which faithfully recapitulates the functions of the *cis* and *trans* regulatory elements/factors involved in *GCN4* translational control[24,25,30–37]. This reporter contains the entire *GCN4* transcription unit, including the ~600 nt leader harboring the four uORFs (Fig. 2), with the *lacZ* coding sequences inserted in-frame with the *GCN4* main ORF. To study uORF1, we employed the "uORF1-only" reporter lacking the AUGs of uORFs 2–4. As it was shown previously that leaky-scanning of the WT uORF1 AUG codon is extremely infrequent[36], expression of β-galactosidase from this reporter provides a read-out of REI following termination at uORF1. Similarly, uORF4 variants were examined in the "uORF4-only" reporter lacking the AUGs of uORFs 1–3, which produces low levels of β-galactosidase owing to a lack of leaky-scanning of the uORF4 AUG codon in the absence of the other uORFs and highly inefficient REI downstream following uORF4 translation[34]. Both uORF1 and uORF4 are present at their normal positions in the leader of these reporters.

We first examined the relative contribution of the Tma20/Tma22 heterodimer and Tma64 to the efficiency of REI following uORF4 translation, which we regard as an exemplar of typical REI-non-permissive uORFs in budding yeast[3,13] (Fig. 1B(i), (ii)). To this end, uORF4-only reporters were generated (Fig. 3A) with the native penultimate CCG^Pro

codon replaced by either the TTG^Leu codon or TGG^Trp codon, judged to be the most or least dependent on Tma factors, respectively, for bulk 40S recycling in yeast[16]. (Tma-dependence for each triplet was assigned previously as the ratio of 40S occupancies in *tma20Δtma64Δ* vs. WT cells averaged across the stop codons of all genes. Here, we designate penultimate codons as being "hyperdependent" or "hypodependent" on Tma factors for recycling if they showed, respectively, greater or less than average *tmaΔΔ*/WT 40S ratios at a 99% confidence level; with all other triplets regarded as displaying "average" Tma-dependency[16].) Assaying the reporters in the three single deletion strains lacking *TMA20*, *TMA22*, or *TMA64*, the *tma20Δtma22Δtma64Δ* triple deletion mutant, and the isogenic WT revealed that the *tma20Δ* and t*ma22Δ* single deletion strains, and the triple deletion mutant (*tmaΔΔΔ*) produced a very modest, albeit significant change in expression of the reporter with the Tma-hypodependent TGG^Trp codon (Fig. 3B). This result matches the expected outcome depicted in Fig. 1B(ii). At odds with the expectations in Fig. 1B(i), however, the single deletions of *TMA20* or *TMA22* conferred significant ≥2-fold derepression of the reporter containing the Tma-hyperdependent TTG^Leu codon. Although the single deletion of *TMA64* had only a small effect of lower significance, the triple mutant showed even higher expression of the TTG^Leu reporter compared to the *tma20Δ* and *tma22Δ* single mutants, at >3-fold above the level in WT (Fig. 3B). One way to explain these unexpected findings on the TTG^Leu reporter is to propose that eliminating Tma20/Tma22's second function in dissociating post-termination 40S subunits from mRNA allows the retained subunits to traverse the leader and reinitiate downstream at the *GCN4* AUG codon, even though the first function of the heterodimer in releasing the deacylated leucyl tRNA is absent in *tmaΔ* mutant cells. As elaborated below, this in turn implies the occurrence of Tma-independent release of the deacylated tRNA at the TTG^Leu codon, allowing increased REI to occur when dissociation of the post-termination 40S subunit is impaired by *tma20Δ* or *tma22Δ* mutations. This proposal of both Tma-dependent and -independent mechanisms of tRNA release at uORF4 variants harboring Tma-hyperdependent penultimate codons is supported by additional evidence presented below.

The results on the TTG^Leu reporter in Fig. 3B further suggest that although Tma64 cannot fully substitute for the Tma20/Tma22 heterodimer it still contributes to inhibiting REI in cells lacking the heterodimer, as eliminating all three Tma proteins in the triple mutant conferred significantly greater derepression of the TTG^Leu reporter compared to the *tma20Δ* or *tma22Δ* single mutants (Fig. 3B). This inference was confirmed by our finding that the triple mutant and *tma20Δtma64Δ* double mutant, in which both the heterodimer and Tma64 are missing, exhibit significantly greater derepression of the TTG^Leu reporter compared to the *tma20Δtma22Δ* double mutant still containing Tma64 (Fig. 3C). Consistent

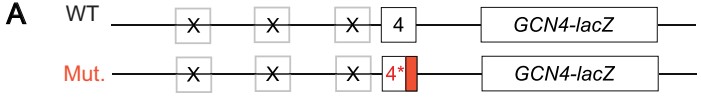

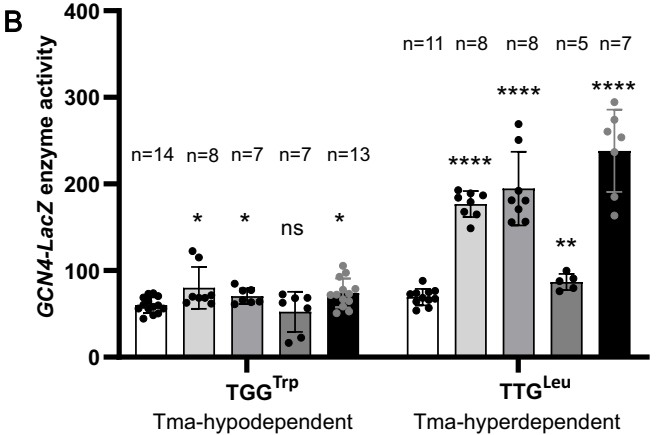

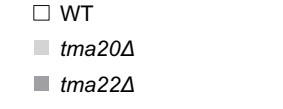

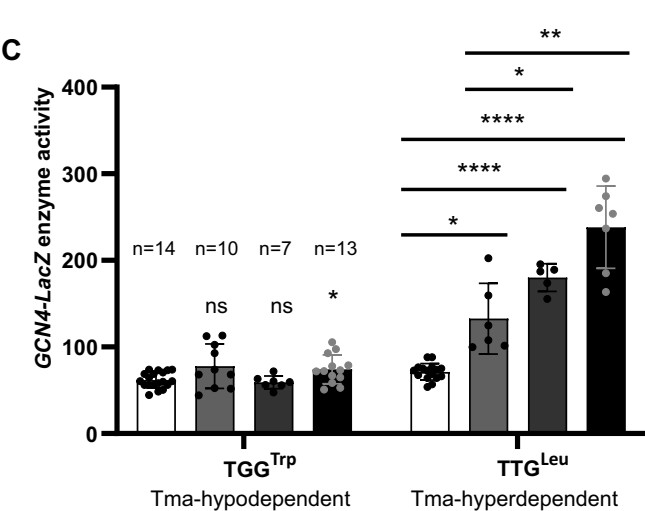

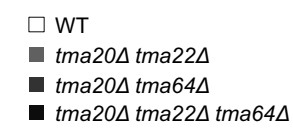

with this, no discernible difference in growth rate was observed between the *tma20Δtma64Δ* double mutant and the triple mutant, both of which grow slightly more slowly than the WT (Fig. 3D). Accordingly, in subsequent experiments, we considered the *tma20Δtma64Δ* and *tma20Δtma22Δtma64Δ* strains to be equivalent in completely lacking the functions of both the heterodimer and Tma64 in regulating REI.

## Tma proteins suppress REI at uORF4 variants containing Tma-hypodependent as well as Tma-hyperdependent penultimate codons

To extend our analysis of the impact of different penultimate codons on Tma-dependent 40S recycling and REI, we examined uORF4-only reporters carrying the three classes of penultimate codons listed in Table 1 that are

**Fig. 3 | Differential effects of various *TMA* deletions on expression of uORF4-only *GCN4-lacZ* reporters equipped with Tma-hypodependent or Tma-hyperdependent penultimate codons. A** Schematic of the uORF4-only *GCN4-lacZ* reporters harboring either WT or mutant penultimate codons. **B** Yeast strains YSG181 (*tma20Δ*), YSG184 (*tma22Δ*), YSG178 (*tma64Δ*), or YKJ3 (*tma20Δt-ma22Δtma64Δ*), deleted for one or all three *TMA* genes and the corresponding WT strain YSG142 (WT BY4741) were transformed with uORF4-only reporters containing WT uORF4 (p226) or uORF4 variants with the penultimate Tma-average codon CCG exchanged for Tma-hyperdependent TTG^Leu (pKJ34) or Tma-hypodependent TGG^Trp triplet (pKJ36). Reporter activity was assayed in whole cell extracts for at least five independent transformants (with the specific *n* denoted in each graph) and activities in the *tmaΔ* transformants were normalized as described in "Methods". Data are represented as ratios of normalized mean values ± SD of β-galactosidase activities in *tmaΔ* strains/WT strain. *$p < 0.05$, **$p < 0.01$, ***$p < 0.001$, ****$p < 0.0001$, ns not significant; for details of statistical analysis, see "Materials and methods". The exact *p* values can be found in Supplementary Data 1. **C** Same analysis as in (**B**) except comparing transformants of the double mutants YKJ6 (*tma20Δtma22Δ*) and YSG196 (*tma20Δtma64Δ*) to the triple mutant YKJ3 (*tma20Δtma22Δtma64Δ*). **D** Serial dilutions of strains of the indicated genotypes from (**C**) were spotted on minimal SD medium and grown for 48 h at 30 °C.

## Table 1 | Fold-changes in uORF4-only and uORF1-only reporter expression in *tma* mutant vs. WT strains[a]

| Penultimate codon | *tma20Δ tma22Δ*/WT ratio (uORF4) ± SD | *tmaΔΔΔ*/WT ratio (uORF4) ± SD | *tma20Δ tma64Δ*/WT ratio (uORF1) ± SEM |
|---|---|---|---|
| **Tma-hypodependent** | | | |
| TAC^Tyr | 2.33 ± 0.64 | 1.47 ± 0.19 | 0.73 ± 0.24 |
| TGG^Trp | 1.30 ± 0.43 | 1.23 ± 0.28 | 1.15 ± 0.58 |
| CAA^Gln | 1.42 ± 0.07 | 2.48 ± 0.45 | 1.06 ± 0.21 |
| GCT^Ala | 2.10 ± 0.23 | 2.83 ± 0.57 | 1.04 ± 0.12 |
| GAC^Asp | N/A | N/A | 1.03 ± 0.13 |
| GAG^Glu | N/A | N/A | 0.92 ± 0.02 |
| CAC^His | N/A | N/A | 0.84 ± 0.17 |
| TTT^Phe | N/A | N/A | 0.94 ± 0.32 |
| TGT^Cys | N/A | N/A | 0.85 ± 0.10 |
| **Tma-average** | | | |
| CCG^Pro (uORF4 WT) | 1.50 ± 0.53 | 1.67 ± 0.53 | 0.92 ± 0.11 |
| GCG^Ala | 2.92 ± 0.07 | 1.55 ± 0.19 | 0.79 ± 0.03 |
| CTG^Leu | 4.13 ± 1.42 | 1.59 ± 0.34 | 0.76 ± 0.09 |
| ATG^Met | 2.55 ± 0.33 | 2.73 ± 0.39 | 0.40 ± 0.03 |
| CCA^Pro | 2.46 ± 0.11 | 3.13 ± 0.55 | 0.44 ± 0.16 |
| TGC^Cys (uORF1 WT) | N/A | N/A | 0.78 ± 0.17 |
| GAA^Glu | N/A | N/A | 0.87 ± 0.07 |
| GTC^Val | N/A | N/A | 0.69 ± 0.04 |
| GGG^Gly | N/A | N/A | 1.01 ± 0.13 |
| CGG^Arg | N/A | N/A | 0.65 ± 0.19 |
| CCC^Pro | N/A | N/A | 0.91 ± 0.18 |
| TCG^Ser | N/A | N/A | 0.84 ± 0.05 |
| AGG^Arg | N/A | N/A | 1.01 ± 0.12 |
| CGC^Arg | N/A | N/A | 0.67 ± 0.20 |
| **Tma-hyperdependent** | | | |
| TTG^Leu | 1.91 ± 0.58 | 3.43 ± 0.47 | 0.59 ± 0.06 |
| ATT^Ile | 3.36 ± 1.26 | 2.40 ± 0.55 | 0.85 ± 0.33 |
| TAT^Tyr | 1.55 ± 0.33 | 2.92 ± 0.36 | 0.96 ± 0.05 |
| AAT^Asn | 2.26 ± 0.70 | 4.56 ± 0.78 | 0.83 ± 0.13 |
| AAA^Lys | N/A | N/A | 0.76 ± 0.11 |
| AAG^Lys | N/A | N/A | 0.86 ± 0.07 |

*N/A* not available.

[a]List of the used constructs classified according to the Tma-dependency of their penultimate codons according to Young et al.[16]. This table contains average fold-change ± SD in reporter expression from biological replicates in either *tma20Δ tma22Δ tma64Δ* strain (YKJ3) and *tma20Δ tma22Δ* strain (YKJ6) (uORF4) or *tma20Δ tma64Δ* (H4520) (uORF1) vs. corresponding WT.

Tma-hypodependent (Fig. 4B), Tma-average (Fig. 4C), or Tma-hyperdependent (Fig. 4D) for 40S recycling in the translatome[16]. To this end, we measured their expression in both the *tma20Δtma22Δ* double mutant and the triple mutant to provide insights into the codon dependence of Tma64 vs. the Tma20/Tma22 heterodimer in suppressing REI. Considering first results for the *tmaΔΔΔ* triple mutant, we observed that all of the reporters with Tma-hypodependent codons (Fig. 4B, black bars) or Tma-average codons (Fig. 4C, black bars) showed statistically significant derepression in the triple mutant, ranging from 1.2-fold to 3.1-fold for different codons (Table 1, Tma-hypodependent & Tma-average data in col. 3). Derepression was also observed for the four reporters with Tma-hyperdependent codons (Fig. 4D; black bars), ranging from 2.4-fold to 4.6-fold (Table 1, Tma-hyperdependent data in col. 3). Thus, strong derepression in the triple mutant was observed for multiple reporters representing Tma-hypodependent codons (CAA^Gln, GCT^Ala) and Tma-average codons (ATG^Met, CCA^Pro) in addition to the four Tma-hyperdependent codons we examined.

Despite this apparent discrepancy with the previously published data on the Tma-dependency of different penultimate codons[16], plotting the expression changes for Tma-hyperdependent, Tma-hypodependent or Tma-average penultimate codons measured in the triple mutant vs. WT (Fig. 5A) revealed a significant tendency for the four Tma-hyperdependent codons to exhibit greater derepression in the triple mutant compared to the four Tma-hypodependent codons tested ($p = 0.0404$). Nonetheless, since the Tma-average reporters containing ATG^Met or CCA^Pro and the Tma-hypodependent GCT^Ala and CAA^Gln reporters did not exhibit statistically smaller derepression ratios in the triple mutant compared to the Tma-hyperdependent reporters for codons TTG^Leu, ATT^Ile, and TAT^Tyr (Table 1, col. 3), we conclude that the Tma-dependence of penultimate codons determined previously[16] is not a strong indicator of Tma-mediated suppression of REI following uORF4 translation.

Considering next results obtained in the *tma20Δtma22Δ* double mutant, where Tma64 can potentially substitute for the heterodimer, we found that five of the uORF4-only reporters that showed marked derepression in the triple mutant displayed substantially less derepression in the *tma20Δtma22Δ* double mutant, including reporters with CAA^Gln and GCT^Ala (Fig. 4B), CCA^Pro (Fig. 4C), TTG^Leu, TAT^Tyr, and AAT^Asn (Fig. 4D) penultimate codons (summarized in Table 1, cf. cols. 2–3). For these six reporters, containing Tma-hypodependent, -average, or -hyperdependent codons, it appears that Tma64 can partially substitute for heterodimer function in suppressing REI. The Tma-hypodependent CAA^Gln (Fig. 4B) and Tma-hyperdependent TAT^Tyr (Fig. 4D) codons illustrate the most complete functional substitution by Tma64, with only slight derepression in the *tma20Δtma22Δ* double mutant despite marked derepression in the triple mutant. Reporters with TAC^Tyr (Tma-hypodependent; Fig. 4B), ATG^Met, CTG^Leu and GCG^Ala (Tma-average; Fig. 4C), and ATT^Ile (Tma-hyperdependent; Fig. 4D) codons appear to represent the other extreme, wherein Tma64 cannot functionally compensate at all and eliminating the heterodimer alone in the double mutant confers derepression equal to or even greater than that found in the triple mutant also lacking Tma64. Finally, the comparatively low-level derepression observed in both mutants for the native uORF4 CCG^Pro (Fig. 4C) and TGG^Trp reporters (Fig. 4B, black bars) makes it difficult to determine whether Tma64 can partially substitute

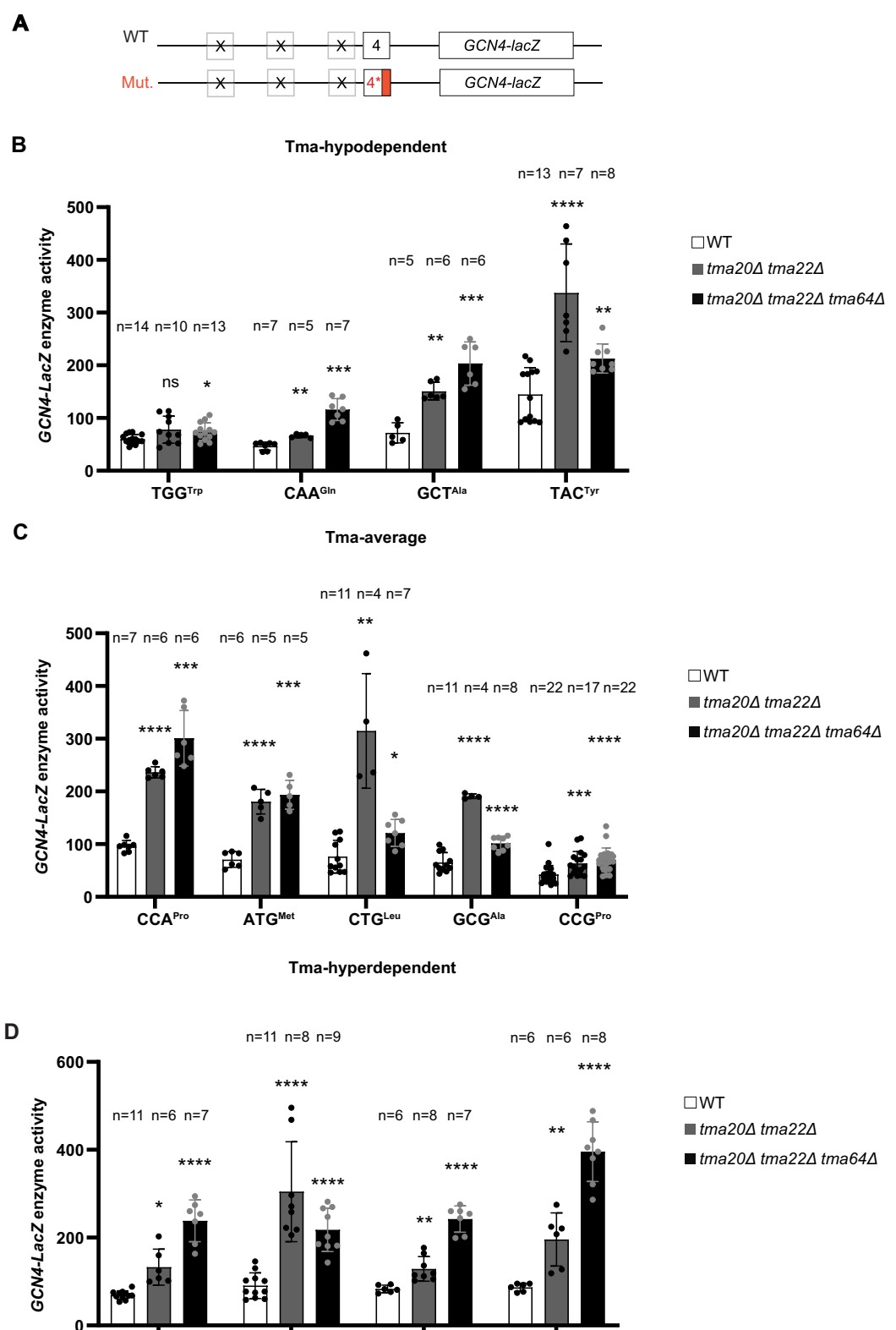

**Fig. 4 | Differential effects of various *TMA* double and triple deletions on expression of uORF4-only *GCN4-lacZ* reporters equipped with Tma-hypodependent, Tma-average or Tma-hyperdependent penultimate codons.**
**A** Schematic of the uORF4-only *GCN4-lacZ* reporters harboring either WT or mutant penultimate codons. **B–D** Yeast strains YKJ6 (*tma20Δtma22Δ*) and YKJ3 (*tma20Δtma22Δtma64Δ*) and the corresponding WT strain were transformed with the WT uORF4-only *GCN4-lacZ* reporter construct (p226) and variants with the WT uORF4 penultimate codon (CCG) exchanged for the indicated Tma-hypodependent, Tma-average or Tma-hyperdependent codons. Reporter activity was assayed in whole cell extracts for at least four independent transformants (with the specific *n* denoted in each graph) and activities in the *tmaΔ* transformants were normalized as described in "Methods". Data are represented as *GCN4-LacZ* reporter enzyme activities in WT strain and *tmaΔ* strains. *$p < 0.05$, **$p < 0.01$, ***$p < 0.001$, ****$p < 0.0001$, ns not significant; for details of statistical analysis, see "Materials and methods". The exact *p* values can be found in Supplementary Data 1.

**Fig. 5 | Changes in reporter expression for groups of Tma-hyperdependent, Tma-hypodependent or Tma-average penultimate codons in *tma* mutant vs. WT cells. A** Violin plot of the average fold-changes in expression of uORF4-only *GCN4-lacZ* reporters harboring the four Tma-hyperdependent (Hyperdep), four Tma-hypodependent (Hypodep) or five Tma-average (Average) penultimate codons in the triple deletant yeast strain YKJ3 (*tma20Δtma22Δtma64Δ*) vs. isogenic WT strain. The average fold-change of expression determined from biological replicates only was calculated for each construct from results plotted in Fig. 4. Statistical analysis was performed using two-tailed unpaired *t*-test *$p < 0.05$. **B** Violin plot of fold changes in expression of uORF1-only *GCN4-lacZ* reporters harboring the six Tma-hyperdependent, nine Tma-hypodependent and 14 Tma-average penultimate codons in *tma20Δtma64Δ* strain H4520 vs. WT strain BY4741. The average fold-change of expression determined from biological replicates was calculated for each construct from the results plotted in Fig. 6. Statistical analysis was performed using the two-tailed unpaired *t*-test. *$p < 0.05$. The exact *p* values can be found in Supplementary Data 1.

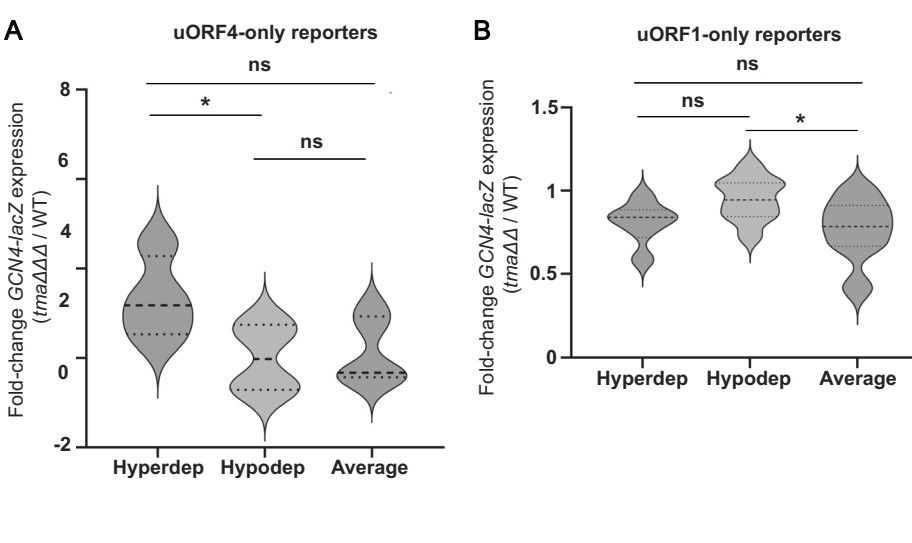

## Very few penultimate codons confer Tma-dependence at REI-permissive *GCN4* uORF1

The mammalian DENR/MCTS1 heterodimer was shown to promote REI after certain short uORFs in mammalian cells, including *ATF4* uORF1[13]; whereas the yeast Tma20/Tma22 counterpart functioned oppositely to inhibit REI following uORF translation in yeast extracts[16]. As mentioned above, it was suggested that the discrepancy between mammals and yeast arises from the inability of post-termination 40S complexes at typical yeast uORFs to remain associated with the mRNA and migrate to downstream start codons following release of the tRNA decoding the penultimate codon[13] (Fig. 1B(i)-(ii) vs. 1A(i)-(ii)). To test this hypothesis, we analyzed the effects of the *tma* mutations on REI at *GCN4* uORF1. As mentioned, this atypical uORF is surrounded by *cis*-acting RPEs, some of which interact with eIF3 subunits and promote retention of post-termination 40S complexes on the mRNA, enabling high-level REI (Fig. 2). Indeed, REI after uORF1 translation is ~20–25-fold higher than after uORF4 translation[25,26,32,33]. The retention of post-termination 40S subunits at uORF1 makes the uORF1-only *GCN4-lacZ* reporter (Fig. 6A) ideally suited to examine whether Tma factors are required to dissociate deacylated tRNA and enable REI at the *GCN4* start codon in a manner limited to penultimate codons with heightened dependence on the Tma factors for bulk 40S recycling in yeast[16]. If so, then uORF1-only reporters equipped with Tma-hyperdependent codons should exhibit reduced REI and *GCN4-lacZ* expression in *tmaΔΔ* vs. WT cells in the manner observed for short uORFs in mammalian cells depleted of DENR (Fig. 1A(i)).

We exchanged the native uORF1 penultimate codon TGC^Cys in the uORF1-only reporter with other penultimate codons of all three classes defined above, including six Tma-hyperdependent, nine Tma-hypodependent, and 13 Tma-average codons. The native TGC^Cys codon is Tma-average, and the native TGT^Cys codon of the REI-permissive uORF2 of

*GCN4*[32] was chosen as one of the Tma-hypodependent codons. Expression of the resulting reporters was assayed in the *tma20Δtma64Δ* double mutant and WT strains (Fig. 6 and Table 1, col. 4). At odds with the prediction that REI would be reduced in the mutant for reporters with Tma-hyperdependent codons, only one of the six reporters of this class, that containing the TTG^Leu codon, showed significantly reduced expression in the mutant (Fig. 6B, hyperdependent constructs). While TTG^Leu was judged to be highly Tma-hyperdependent for bulk 40S recycling, ATT^Ile and TAT^Tyr were equally so[16], yet the reporters for the latter two codons showed unchanged expression in the mutant strain. Moreover, the constructs containing the Tma-average ATG^Met or CCA^Pro codons were the only reporters besides the TTG^Leu construct showing significantly reduced expression in *tma20Δtma64Δ* cells (Fig. 6B, Tma-average constructs). Fold changes in expression between WT and *tmaΔΔ* strains can be found summarized in Table 1, col. 4. Plotting the expression changes for the entire set of 29 codons examined at uORF1 (Fig. 5B) revealed a general tendency for reduced expression in the *tmaΔΔ* mutant vs. WT cells, but no significant difference in median expression ratios between Tma-hyperdependent reporters and either Tma-hypodependent or Tma-average reporters.

Figure 5B does reveal a small but significant difference between the median expression ratios between the Tma-hypodependent and Tma-average groups ($p = 0.0394$), which reflects the slightly increased REI observed in *tmaΔΔ* cells for the Tma-hypodependent group. None of the uORF1-only reporters, however, showed the ≥2-fold increased REI conferred by the *tmaΔΔΔ* mutation for most of the uORF4-only reporters (Fig. 5A), even for the ten cases where uORF1-only and uORF4-only reporters share the same penultimate codons (ATT, TTG, AAT, TAT, ATG, GCG, CTG, CAA, GCT, CCA). This is consistent with the prediction that the specialized RPEs at uORF1 allow 40S post-termination complexes to resist ribosome dissociation by the Tma factors that suppress REI at uORF4 lacking RPEs (Fig. 2). In summary, our findings indicate that uORF1 differs from mammalian short uORFs, including the REI-permissive uORF1 of *ATF4*, in lacking a requirement for Tma factors in dissociating deacylated tRNA from post-termination complexes at penultimate codons that impose a heightened Tma-requirement in bulk 40S recycling. A lack of Tma-dependence on REI also applies to eight of the nine DENR-dependent codons[13] represented among our uORF1-only reporters (ATT, GCG, CTG, CCG, CGG, CCC, AGG, and GCT). In fact, ATG is the sole DENR-dependent codon we tested that confers a dependence on Tma factors for

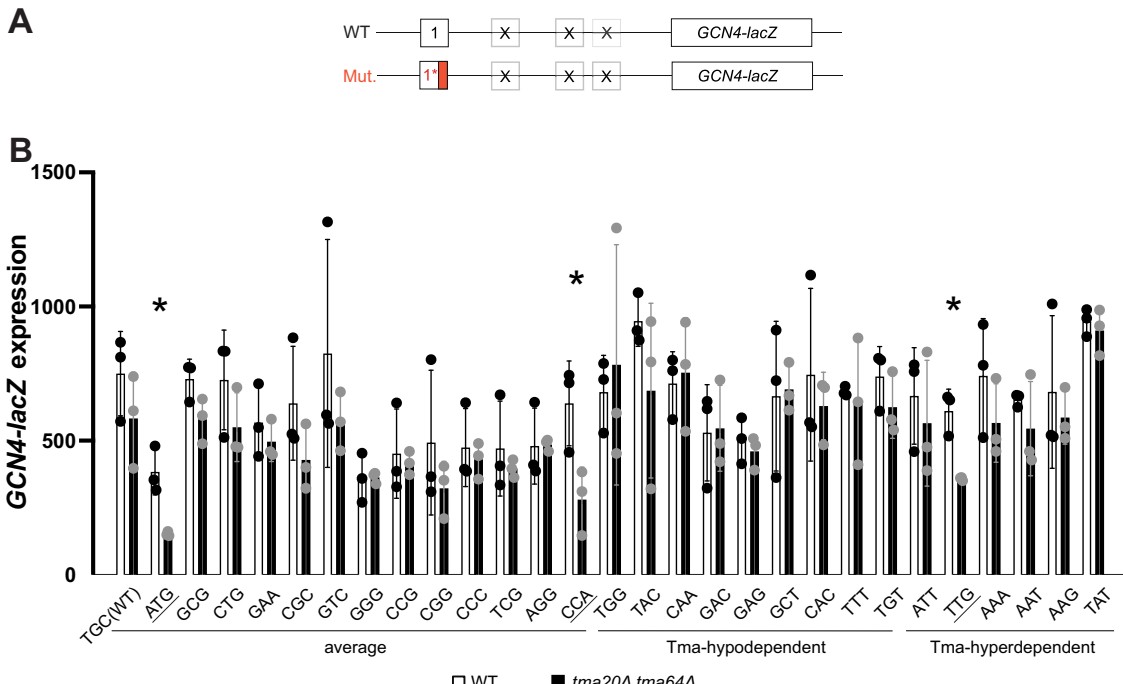

**Fig. 6 | Differential effects of deleting both *TMA20* and *TMA64* on expression of uORF1-only *GCN4-lacZ* reporters equipped with Tma-average, Tma-hypodependent or Tma-hyperdependent penultimate codons. A** Schematic of the uORF1-only *GCN4-lacZ* reporters harboring either WT or mutant penultimate codons. **B** Yeast strains H4520 (*tma20Δtma64Δ*) and WT strain BY4741 were transformed with uORF1-only *GCN4-lacZ* reporter constructs with the WT uORF1 penultimate codon (TGC) exchanged for the indicated Tma-average, Tma-hypodependent or Tma-hyperdependent codons. β-galactosidase activities determined for each of three independent transformants (*n* = 3) of the *tma20Δtma64Δ* strain were divided by a factor of 1.316 prior to calculating the mean activities and S.E.M. values plotted here to normalize for differences in reporter expression between the two strains that are independent of the uORF1 variants and observed for an uORF-less reporter (as described in "Methods"). The mean values between mutant and WT were compared for each reporter in a two-tailed, unpaired Student's *t* test to assign *p* values (**p* < 0.05). The exact *p* values can be found in Supplementary Data 1.

REI at uORF1, but as noted above, ATG is Tma-average for 40S recycling in yeast.

In summary, neither Tma-dependence nor DENR-dependence of the penultimate codon is an accurate predictor of Tma-dependence for REI at *GCN4* uORF1 variants, despite the functional similarity between *GCN4* uORF1 and *ATF4* uORF1.

**Function of yeast "start-stop" elements is not Tma-dependent**
As mentioned earlier, mammalian DENR was shown to be required for REI after start-stop uORFs consisting of ATG followed by a stop codon[13,18]. To test the Tma-dependence of start-stops for downstream REI in yeast, we deleted the 2nd and 3rd codons from native uORF1-, uORF3- and uORF4-only constructs, without altering any other sequences, producing uStart-stop-only constructs uSt-st1, uSt-st3 and uSt-st4, respectively. (See bottom of Fig. 2 for sequences of the three uSt-st uORFs.)

First, we noticed that all three start-stop elements are more inhibitory to REI than their native uORF counterparts, significantly reducing *GCN4-lacZ* expression in WT cells by ~50-fold, ~3-fold, and ~0.6-fold compared to the levels observed for native uORF1-only, uORF3-only, and uORF4-only reporters, respectively (Fig. 7A). Based on findings in mammalian cells, we reasoned that converting native uORF1 to uSt-st1 would confer a dependence on REI by the Tma proteins not observed for native uORF1 with its penultimate codon TGC^Cys (Fig. 6, TGC(WT)), reducing expression in *tmaΔΔ* cells. This was not observed, however (Fig. 7B, cols. 1–4), even though we found above that ATG^Met as the penultimate codon for 3-codon uORF1 conferred Tma-dependence for efficient REI (Fig. 6, ATG). Interestingly, converting uORF4 to uSt-st4 eliminated the significant derepression observed in *tmaΔΔ* cells for the native uORF4-only reporter (Fig. 7B, cols. 9–12). This suggests that conversion to a start-stop element eliminates the requirement for Tma factors for highly efficient 40S recycling at uORF4 and repression of REI downstream. The Tma proteins appear to make no

contribution to recycling and REI at native uORF3, as the uORF3-only reporter expression is unaffected by the *tma20Δtma64Δ* mutations; whereas its conversion to uSt-st3 confers a modestly significant Tma-dependence for recycling and suppression of REI (Fig. 7B, cols. 5–8). Overall, our findings suggest that, unlike their mammalian counterparts, the Tma proteins play little role in controlling REI after start-stop elements in yeast.

## Discussion
As yeast Tma20/Tma22 and, to a lesser degree, Tma64 have been implicated in 40S ribosome recycling in vivo[15,16], we used an established reporter system based on *GCN4* translational control to examine the influence of yeast Tma20/MCTS1, Tma22/DENR and Tma64/eIF2D on REI downstream of short uORFs optimized for either REI (uORF1) or for recycling and suppression of REI (uORF4)[5]. In addition, we dissected the functional interplay between the Tma factors in regulating these alternative post-termination outcomes. To do so, we employed variants of uORF1 or uORF4 containing various penultimate codons that conferred either unusually strong dependence, average, or lower than average dependence on Tma proteins for bulk 40S recycling at most stop codons in yeast as genetic tools.

Using the uORF4-only *GCN4-lacZ* reporter, we first demonstrated that inserting Tma-hyperdependent codon TTG^Leu substantially increased REI in single mutants lacking *TMA20* or *TMA22* but only slightly in the mutant lacking *TMA64* (Fig. 3A). Deleting *TMA64* in the *tma20Δtma22Δ* mutant significantly enhanced this defect (Fig. 3C), suggesting that Tma64 can partially compensate for the absence of Tma20/Tma22 in suppressing REI. This conclusion agrees with previous 40S profiling data[16], wherein a single deletion of *TMA20* or *TMA22*, but not *TMA64*, led to accumulation of unrecycled 40S subunits at stop codons translatome-wide, and this phenotype was more pronounced in the *tma20Δtma64Δ* deletion strain lacking both the heterodimer and Tma64. Consistent with this, we observed significantly increased REI for the TTG^Leu uORF4 variant in the

**Fig. 7 | Effects of start-stop elements on downstream REI in the *GCN4-lacZ* reporter system and differential effects of *TMA20* and *TMA64* deletion on re-initiation downstream of start-stop elements. A** WT strain BY4741 was transformed with *GCN4-lacZ* reporter plasmids containing the following single WT uORFs: uORF1-only (p209), uORF3-only (pSG61_(2)), or uORF4-only (p226); or containing single Start-stop elements uSt-st1 (pKP78), uSt-st3 (pKP76), or uSt-st4 (pKP77). β-galactosidase activities were assayed in at least four biological replicates (with the specific *n* denoted in each graph) and data are presented as *GCN4-LacZ* reporter enzyme activities. **B** Strains YSG196 (*tma20Δ tma64Δ*) and the corresponding WT were transformed with the same set of *GCN4-lacZ* reporter plasmids described in (**A**) and reporter expression was measured as described there. Reporter activity values in the *tmaΔΔ* strain were normalized as described in "Methods". *$p < 0.05$, **$p < 0.01$, ***$p < 0.001$, ****$p < 0.0001$, ns not significant; for details of statistical analysis, see "Materials and methods". The exact *p* values can be found in Supplementary Data 1.

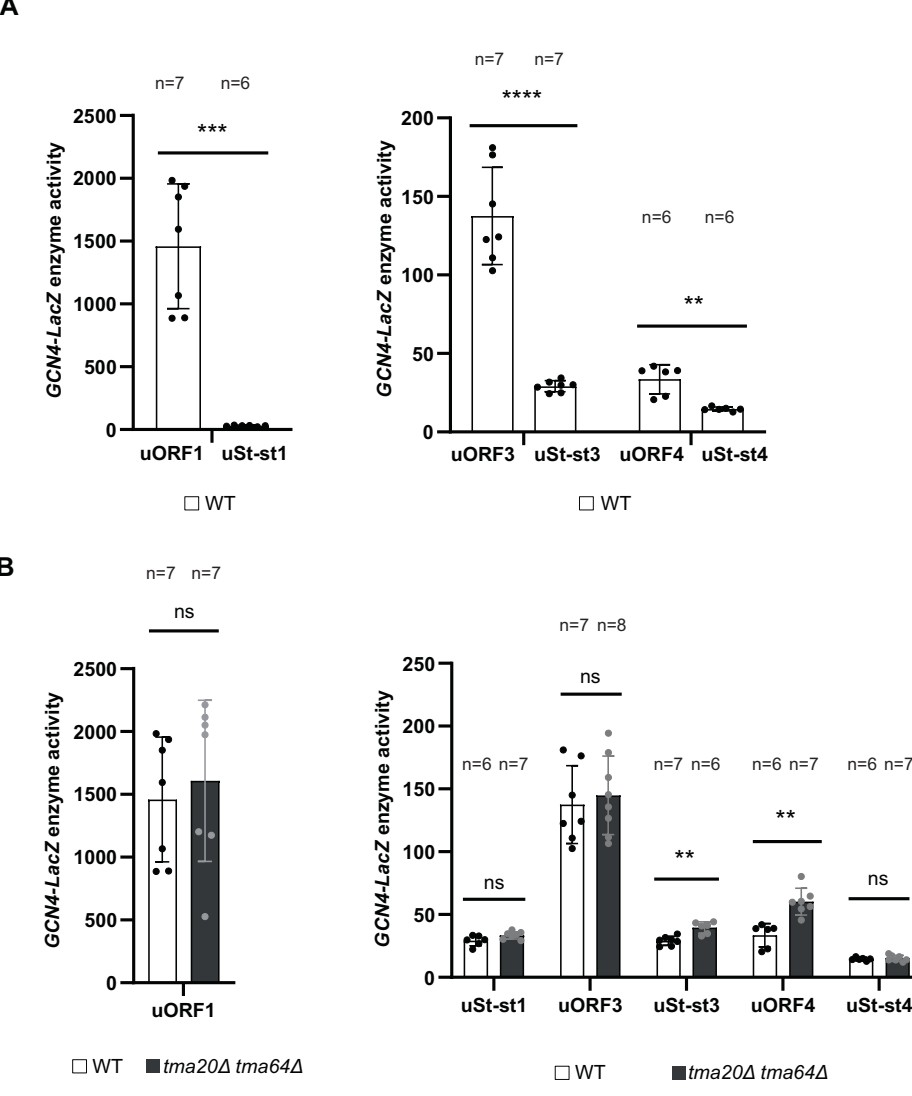

*tma20Δtma64Δ* double mutant and *tma20Δtma22Δtma64Δ* triple mutant compared to the *tma20Δtma22Δ* double mutant (Fig. 3C). We concluded that intact Tma20/Tma22 heterodimer is indispensable for WT 40S recycling and suppression of REI but receives functional support from Tma64 at the stop codon of this uORF4 variant.

Examining a total of 12 different uORF4 variants with different penultimate codons, plus native uORF4, revealed that the absence of all three Tma proteins in the *tmaΔΔΔ* mutant increased REI by less than 2-fold for native uORF4 containing CCG^Pro, and for the TAC^Tyr, CTG^Leu, GCG^Ala and TGG^Trp uORF4 variants, but by more than 2-fold for the CAA^Gln, GCT^Ala, CCA^Pro and ATG^Met variants (Fig. 4B, C and Table 1). Since all these codons showed average or hypodependency on Tma factors in bulk recycling[16], these mixed outcomes indicate that the Tma-dependency in bulk 40S recycling determined previously[16] is generally a weak predictor of Tma-dependence in promoting 40S recycling and suppressing REI for different penultimate codons at uORF4. On the other hand, our finding that the group of Tma-hyperdependent reporters shows a significantly greater increase in median expression compared to the Tma-hypodependent and Tma-average groups (Figs. 4D, 5A and Table 1) suggests that Tma-hyperdependence in bulk recycling at least partly contributes to the impact of Tma proteins in blocking REI observed for uORF4 variants encompassing a range of Tma-dependencies.

Our finding that REI is elevated in *tmaΔΔΔ* cells at uORF4 variants containing Tma-average or Tma-hypodependent codons is difficult to explain if dissociation of the empty 40S subunit is believed to occur spontaneously at these penultimate codons, which should maintain low-level REI in the presence or absence of Tma factors (Fig. 8A(ii)-(a)). Hence, in the model proposed in Fig. 8A(ii)-(b), we assume that Tma factors must figure prominently in dissociating the empty 40S subunits at Tma-hypodependent codons (recycling step 2 in WT), such that retention of a fraction of 40S subunits in their absence in *tmaΔΔ* cells allows increased REI downstream when release of the deacylated tRNA is achieved by a Tma-independent pathway. Our observation that the Tma proteins also inhibit REI at uORF4 variants containing Tma-hyperdependent codons is likewise difficult to explain. Specifically, the failure to release the deacylated tRNA in the *tmaΔΔ* mutant should block REI even though Tma-catalyzed dissociation of the 40S subunit from mRNA is absent, thus keeping REI low in *tmaΔΔ* cells (Fig. 8A(i)-(a)). Therefore, it seems necessary to propose that the deacylated tRNA eventually dissociates spontaneously or by another pathway in the absence of Tma factors, so that loss of Tma-catalyzed dissociation of the empty 40S subunit can enable REI and confer the increased REI we observed in *tmaΔΔ* cells (Fig. 8A(i)-(b)).

The models we propose in (b) panels of Fig. 8A(i)-(ii) assume that Tma-independent recycling can operate at all stop codons and that recycling at Tma-hyperdependent penultimate codons simply relies more heavily on Tma factors than at Tma-hypodependent codons. This possibility is consistent with the fact that eliminating all three Tma proteins confers only a

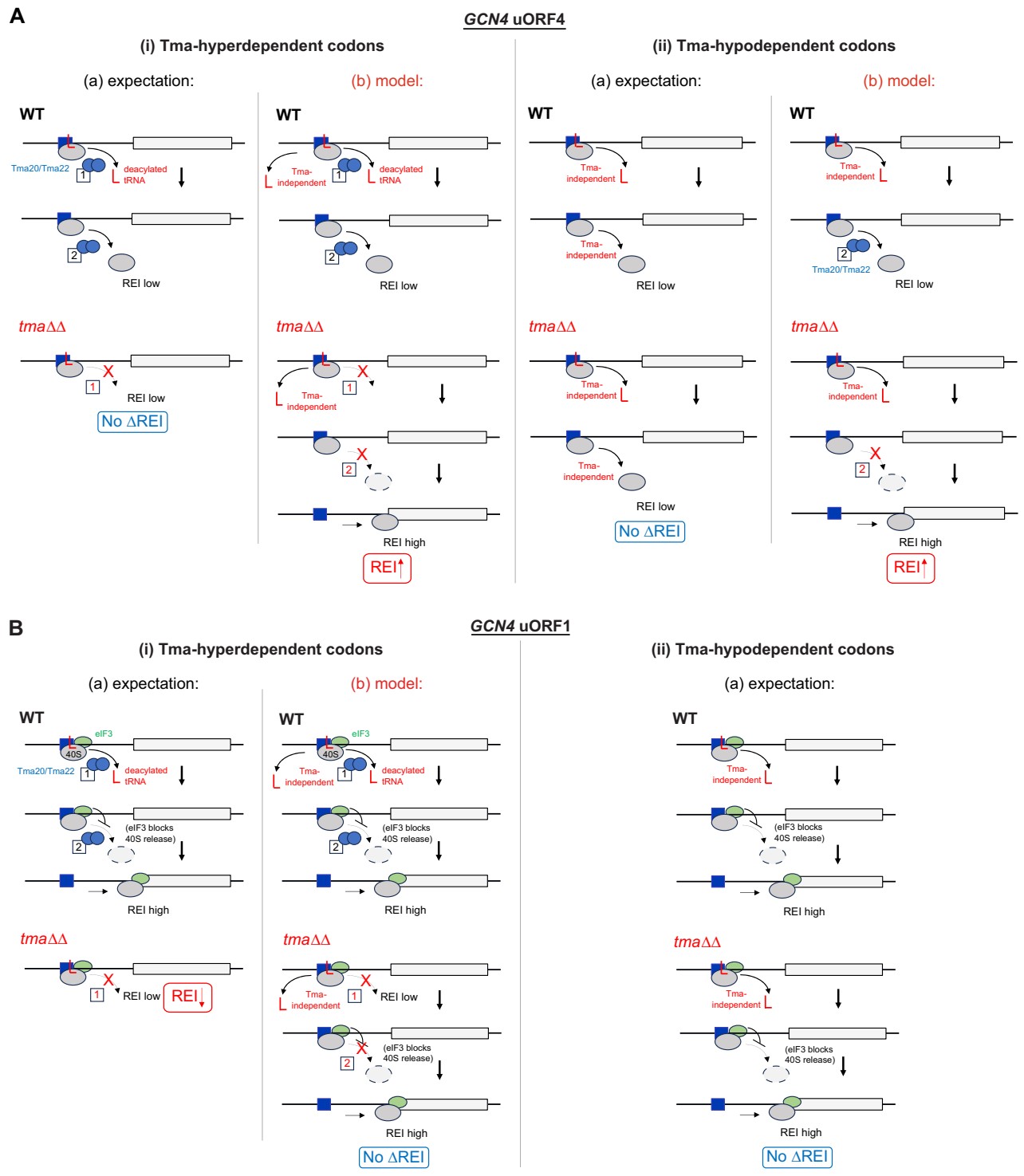

small effect on cell growth, implying the frequent occurrence of Tma-independent recycling throughout the translatome.

We also investigated whether the Tma factors function similarly to MCTS1/DENR in promoting, rather than impeding, REI at short uORFs that are permissive for REI owing to retention of eIFs during uORF translation. It was proposed that the persistence of eIFs on 40S post-termination complexes prevents dissociation of the 40S subunits following MCTS1/DENR-catalyzed release of deacylated tRNA, allowing resumption of 40S migration and rebinding of TC for REI downstream. This explained why REI was impaired on depletion of MCTS1/DENR only for uORFs containing DENR-dependent penultimate codons[13] (Fig. 1A(i)-(ii)). To

investigate this mechanism in yeast, we examined the *GCN4* uORF1-only reporter in which RPEs surrounding uORF1 increase eIF3 and eIF4F occupancy of the post-termination 40S subunits and confer high-frequency REI at *GCN4* (Fig. 2)[25,26,32,33]. Consistent with results for MCTS1/DENR, REI was not reduced in the *tma*ΔΔ mutant for native uORF1 nor for 21 other uORF1 variants equipped with Tma-average or Tma-hypodependent penultimate codons but REI was reduced when uORF1 contained the Tma-hyperdependent TTG[Leu] codon. However, at odds with results for MCTS1/DENR, REI was not reduced in *tma*ΔΔ cells at five other uORF1 variants with Tma-hyperdependent codons. This included the TAT codon present in uORF1 of *ATF4* mRNA—functionally equivalent to *GCN4* uORF1[38]—

**Fig. 8 | Expected vs. proposed functions for yeast Tma20/Tma22 at REI-permissive *GCN4* uORF1 and REI-non-permissive *GCN4* uORF4.** Tma20/Tma22 (blue ovals), penultimate deacylated tRNA (red "L"), and the two functions of Tma factors (boxes 1 and 2) in releasing tRNA and dissociating empty 40S subunits, respectively (solid or dashed arrows), are depicted as in Fig. 1. **A** Tma-mediated recycling at *GCN4* uORF4 depending on the presence of (i) Tma-hyperdependent or (ii) Tma-hypodependent penultimate codons. (i) Tma-hyperdependent uORFs. (a) expectation: as depicted exactly as in Fig. 1B(i) (presented again for comparison), in WT cells (upper), Tma factors efficiently release the penultimate-codon tRNA and also dissociate the empty 40S subunits, conferring low-level REI. In *tmaΔΔ* cells (lower), the absence of Tma-mediated tRNA release helps to ensure low-level REI, yielding no change in REI in *tmaΔΔ* vs. WT cells. (b) model: in WT (upper), deacylated tRNA can be released by Tma-hyperdependent or Tma-independent mechanisms, but Tma-mediated 40S dissociation maintains low-level REI. In *tmaΔΔ* cells (lower), Tma-independent tRNA release coupled with absence of Tma-mediated 40S dissociation confers the increased REI we observed in *tmaΔΔ* vs. WT cells. (ii) Tma-hypodependent uORFs. (a) expectation: as depicted exactly in Fig. 1B(ii), Tma-independent tRNA release occurs in both WT and *tmaΔΔ* cells, but REI is low in both cases because the empty 40S subunits dissociate from the mRNA independently of Tma factors, for no change in REI vs. WT. (b) model: in WT (upper), 40S dissociation is catalyzed by Tma factors even at these Tma-hypodependent uORFs, and absence of this function in *tmaΔΔ* cells, coupled with Tma-independent tRNA release, confers the observed increased REI in *tmaΔΔ* vs. WT cells. **B** Tma-mediated recycling at *GCN4* uORF1 depending on the presence of Tma-hyperdependent (i) or Tma-hypodependent (ii) codons. (i) Tma-hyperdependent uORFs. (a) expectation: as depicted for MCTS1/DENR in Fig. 1A(i), In WT cells (upper), Tma factors efficiently release the penultimate-codon tRNA but its second function in 40S dissociation is impeded by eIF3 (green oval) to enable high-level REI. In *tmaΔΔ* cells (lower), diminished release of the penultimate-codon tRNA lowers REI. (b) model: deacylated tRNA can be released from uORF1 variants by both Tma-hyperdependent and Tma-independent mechanisms in WT yeast (upper) and Tma-independent tRNA release maintains high-level REI in *tmaΔΔ* cells (lower), as 40S dissociation is impeded by eIF3 in both WT and *tmaΔΔ* cells, thus accounting for the unchanged REI we observed in *tmaΔΔ* vs. WT cells. (ii) Tma-hypodependent uORFs. (a) expectation: as depicted for MCTS1/DENR in Fig. 1A(ii), Tma-independent release of the tRNA and subsequent high-level REI occurs in the presence (top) or absence (bottom) of Tma factors, yielding no decrease in REI in *tmaΔΔ* vs. WT cells, as we observed.

---

that confers DENR-dependent REI on *ATF4* mRNA[13]. Moreover, REI was markedly reduced in *tmaΔΔ* cells when uORF1 contained the Tma-average penultimate codons ATG and CCA. Thus, dependence on the Tma factors for REI was conferred on uORF1 by only three of 29 penultimate codons examined, only one of which (TTG) is Tma-hyperdependent for bulk 40S recycling[16]. Moreover, there was no significant difference in median *tmaΔΔ* derepression ratios between uORF1-only reporters containing Tma-hyperdependent vs. Tma-average or Tma-hypodependent codons (Fig. 5B). While there is a general tendency for REI to be diminished at uORF1 variants in *tmaΔΔ* vs. WT cells (Fig. 5B), the Tma factors do not stimulate REI preferentially at uORF1 variants whose penultimate codons are Tma-hyperdependent in bulk recycling in the manner described for MCTS1/DENR at REI-permissive uORFs in mammalian cells.

One possibility is that the specialized interaction of *GCN4* uORF1 RPEs with eIF3 generally overrides the requirement for Tma factors in dissociating the deacylated tRNA at Tma-hyperdependent penultimate codons. By preventing 40S dissociation from the mRNA, eIF3 might provide sufficient time for dissociation of the deacylated tRNA, either spontaneously or by a Tma-independent pathway, to prevent reduced REI in *tmaΔΔ* cells. This scenario is depicted in the model proposed in Fig. 8B(i)-(b) to explain our unexpected findings on uORF1 variants with Tma-hyperdependent codons. In support, numerous tRNAs were found to readily dissociate from the 40S subunit without the help of any dedicated factor[19]. Furthermore, structural analysis of MCTS1/DENR bound to the 40S subunit indicates that MCTS1 might occupy the same binding site on the ribosome as the eIF3a, b, and c subunits of mammalian eIF3[39]. In addition, the ribosome binding interface in MCTS1 seems to be well conserved from yeast to higher eukaryotes (Fig. 9A), as we also observed when comparing the MCTS1 structure with the Tma20 AlphaFold structure prediction (Fig. 9B). This might indicate competitive ribosome binding between eIF3 and Tma20/Tma22, such that the uORF1 RPEs would promote binding of eIF3a vs. the Tma heterodimer and thereby prevent the latter from influencing REI appreciably at uORF1.

Presumably, interaction of deacylated leucyl tRNA with the TTG^Leu codon is too stable to overcome the requirement for Tma-mediated release, leading to the observed reduced REI for the TTG^Leu uORF1 variant in *tmaΔΔ* cells, in the manner predicted for DENR-hyperdependent codons (Fig. 1A(i)). The same might be true for interaction of elongator methionyl tRNA and prolyl tRNA with their cognate ATG^Met and CCA^Pro codons, which also displayed Tma-dependence for REI at uORF1, even though ATG and CCA were not identified as being Tma-hyperdependent for bulk 40S recycling[16]. Supporting this idea, ATG was identified as a DENR-hyperdependent penultimate codon in mammalian cells both in bulk 40S recycling and reporter REI assays[13].

Another consequence of introducing ATG as the penultimate codon of *GCN4* uORF1 not yet mentioned is a significant reduction in REI in WT cells ($p = 0.04$), which is further exacerbated in the *tmaΔΔ* mutant (Fig. 6). A reduction in WT cells was also found for GGG^Gly ($p = 0.03$), CCG^Pro, CCC^Pro, AGG^Arg and GAG^Glu codon replacements ($p = 0.09$) (Fig. 6B). CCG^Pro is the evolutionarily conserved penultimate codon at uORFs 3 and 4 and was shown previously to inhibit REI when introduced at uORF1[33], which might result from delaying decoding or termination at the adjacent stop codon[33].

In addition to *ATF4* uORF1 and other REI-permissive uORFs, the MCTS1/DENR heterodimer was found to promote REI after translation of the *ATF4* start-stop element preceding uORF1[13] and at various other start-stop elements in optimum Kozak context[13,17,18,20]. In contrast, we found no evidence for Tma-dependent recycling and REI at the start-stop elements we derived from *GCN4* uORFs 1, 3, or 4, all of which contain the optimum A-rich sequence context for initiation in yeast. Converting uORF3 and uORF4 to start-stop elements decreased REI in WT cells and diminished the increased REI observed for native uORF4 in *tmaΔΔ* cells. It appears that uSt-st4 exhibits an elevated level of spontaneous 40S recycling that bypasses the contribution of Tma factors to recycling seen at native uORF4. Converting uORF1 to start-stop element uSt-st1 also dramatically decreased REI in WT cells, apparently disabling the RPEs that enable high-level REI at native uORF1. Although uSt-st1 behaves like a typical inhibitory uORF, 40S recycling appears to occur without any contribution from the Tma factors at this element. The fact that uSt-st1, uSt-st4, and uSt-st3 do not show decreased REI in *tmaΔΔ* cells in the manner we observed for the *GCN4* uORF1 variant with ATG^Met as penultimate codon (Fig. 6B) is not surprising considering that, similar to uORF4, the St-st elements appear to lack the capacity for eIF3-mediated retention of 40S post-termination complexes (Fig. 2).

In summary, we found that most replacements of the uORF4 penultimate codon confer increased REI in mutants lacking both Tma64 and the Tma20/Tma22 heterodimer compared to the native CCG codon, suggesting a codon-dependent role for the Tma factors in dissociating 40S post-termination complexes at this uORF, even for those penultimate triplets that were shown previously to be Tma-hypodependent for recycling at other stop codons throughout the translatome. These findings imply that the deacylated tRNA can dissociate in the absence of Tma factors at both Tma-hyperdependent and Tma-hypodependent penultimate codons, so that loss of Tma-catalyzed recycling of the empty 40S subunit permits elevated REI at uORF4 in the mutant cells (models in (b) panels of Fig. 8A(i)-(ii)). In contrast to findings on short REI-permissive uORFs in mammalian cells, and *ATF4* uORF1 in particular[13], we found that only one of six Tma-hyperdependent penultimate codons introduced at *GCN4* uORF1 conferred reduced REI in *tma20Δtma64Δ* cells. These last findings indicate that

**A**

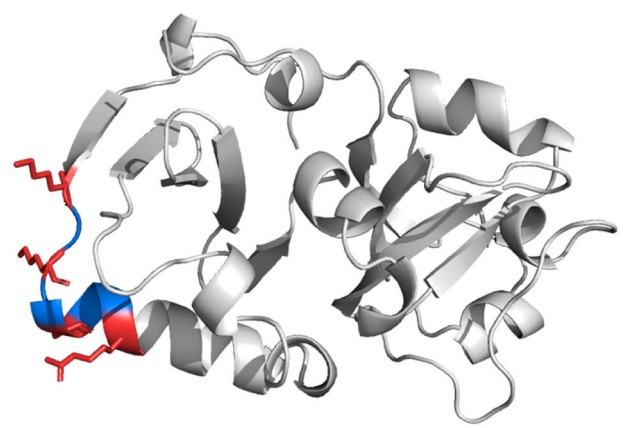

|  |  | Similarity score: | Identity: |
|---|---|---|---|
| *S. cerevisiae* | MFKKFTR-EDVHSRSKVKSSIQRTLKAKLVKQYPKIEDVIDELIPKKSQIELIKCEDKIQ 59 |  |  |
| *D. melanogaster* | MFKKFEEKDSISSIQQLKSSVQKGIRAKLLEAYPKLESHIDLILPKKDSYRIAKCHDHIE 60 | 413 | 49.7% |
| *H. sapiens* | MFKKFDEKENVSNCIQLKTSVIKGIKNQLIEQFPGIEPWLNQIMPKKDPVKIVRCHEHIE 60 | 423 | 48% |
| *C. elegans* | MFKKFDEKEDVTGATQLKSSVQKGIRKKLIENFPYLEPHLEEILPKKENFKVIKCKDHIE 60 | 414 | 46.2% |

**B**

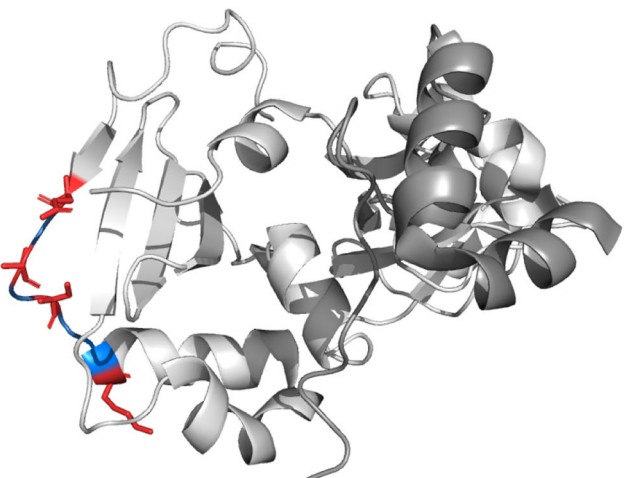

**Fig. 9 | Comparison of *S. cerevisiae* (yeast), *D. melanogaster* (fly), *H. sapiens* (human) and *C. elegans* (worm) Tma20/MCTS1 proteins. A** Multiple sequence alignment was performed using Clustal Omega. This alignment illustrates the conservation of amino acid residues of the ribosome binding interface described previously[39] and marked here by red dots. Similarity score and identity percentage were determined using UniProt BLAST tool. **B** Comparison of predicted Tma20 structure (upper panel) and structure of MCTS1 in complex with DENR (bottom panel; MCTS1 colored in light gray, DENR colored in dark grey). Conserved residues are depicted in red and regions separating them are colored in blue. Tma20 structure was predicted using the AlphaFold3[48] algorithm. MCTS1/DENR complex structure was published previously[39] (PDB accession 6MS4). PyMOL 3.0 was used for data visualization.

dissociation of the deacylated tRNA and subsequent REI can generally proceed without Tma factors at uORF1, even at penultimate codons where these proteins markedly enhance bulk 40S recycling, which we again attribute to Tma-independent tRNA release (model in Fig. 8B(i)-(b)). Together, our results on uORF1 and uORF4 account for our previous finding that *GCN4* translational control was essentially unperturbed in mutants lacking both Tma20/Tma22 and Tma64[29]—in stark contrast to the requirement for MCTS1/DENR for induction of *ATF4* mRNA translation. Also at odds with findings on mammalian MCTS1/DENR, converting uORF1 to a start-stop element did not confer Tma-dependent REI, and converting uORF4 to a start-stop element decreased REI and diminished the contribution of Tma factors in suppressing REI. Overall, the influence of Tma factors on REI at

short regulatory uORFs in yeast cells differs considerably from that observed for mammalian MCTS1/DENR.

## Materials and methods
### Yeast strains and plasmids
The genotypes of all yeast strains employed in this study are listed in Supplementary Table 1. Yeast mutants lacking one or more *TMA* genes were constructed using DNA cassettes bearing *kanMX4*, *hphNT1* or *natNT2* resistance markers contained in plasmids pFA6a-KanMX4[40] (acquired from J.H. Hegeman), pZC3 and pZC4[41], respectively. The cassettes were amplified by PCR using gene-specific primers SG325 and SG326 (*TMA64*/YDR117C) and KJ1 and KJ2 (*TMA22*/YJR014W) (see Supplementary Table 2 for

complete list of primers). Subsequently, the amplified DNA was used to transform yeast to confer resistance to antibiotics Geneticin G418 (Bio-Concept), Nourseothricin (Jena Bioscience), or Hygromycin B (Carl Roth GmbH), respectively, on YPD agar medium. YSG178 (*tma64Δ*) was derived from WT BY4741 (YSG142) by transformation with a *TMA64*-specific *kanMX4* deletion cassette. YSG196 (*tma20Δtma64Δ*) was generated from YSG181 (*tma20Δ*) by transformation with a *TMA64*-specific *natNT2* deletion cassette. YKJ3 (*tma20Δtma22Δtma64Δ*) was generated from YSG196 (*tma20Δtma64Δ*) by transformation with a *TMA22*-specific *hphNT1* cassette. YKJ6 (*tma20Δtma22Δ*) was generated from YSG181 (*tma20Δ*) by transformation with a *TMA22*-specific *natNT2* cassette. Deletion of WT alleles and replacement with the appropriate deletion cassettes was verified by colony PCR[42] using gene-specific primers SG295 (for *TMA20*/YER007C-A), SG294 (for *TMA64*/YDR117C deletions) and SG296 (for *TMA22*/YJR014W deletions) in combination with universal primer PB238 complementary to all deletion cassettes. Strains YSG181 (*tma20Δ*) and YSG184 (*tma22Δ*) were acquired from the Euroscarf yeast deletion collection.

All plasmids employed in this study are listed in Supplementary Data 2. All uORF4-only reporters were created by replacing the *SalI-BstEII* fragment of uORF4-only *GCN4-lacZ* construct p226[43] with the appropriate *SalI-BstEII* fragments generated by either PCR primer mutagenesis or DNA synthesis. Fragments generated by PCR primer mutagenesis were amplified using common forward primer KJ27 (mapping near the *SalI* cloning site) and the following mutagenic reverse primers (mapping near the *BstEII* site) for constructing the indicated plasmids: primer KJ25 for plasmid pKJ34, primer KJ26 for pKJ35, primer KJ24 for pKJ36, primer KJ78 for pKJ57, primer KJ80 for pKJ58, primer KJ76 for pKJ59, primer KJ74 for pKJ61, primer KJ77 for PKJ62. DNA fragments for constructing pKJ17, pKJ23, pKJ20 and pKJ21 were synthesized by GeneArt Gene Synthesis (Thermo Fisher). All fragment inserts in the final constructs were verified by Sanger sequencing, and the DNA sequences are listed in Supplementary Data 3.

The start-stop constructs pKP76, pKP77 and pKP78 were created by replacing the *SalI-BstEII* fragment of the *GCN4-lacZ* construct pSG194[33]; a uORF1-only construct created by replacing the coding sequence of uORF1 by that of uORF2) by the appropriate *SalI-BstEII* fragments generated by GeneArt Gene Synthesis (Thermo Fisher). These fragments contain point mutations in start codons of uORFs (point mutations in uORFs 1, 2 and 4 in pKP76, point mutations in uORFs 1, 2 and 3 in pKP77 and point mutations in uORFs 2, 3 and 4 in pKP78) and deletions of sense codons between start and stop codons in uORF3 (pKP76), uORF4 (pKP77) and uORF1 (pKP78), respectively, which creates start-stop elements uSt-st3, uSt-st4 and uSt-st1. The mutation scheme of these constructs is also shown in Fig. 2. All fragment inserts in the final constructs were verified by Sanger sequencing, and the DNA sequences are listed in Supplementary Data 3.

All uORF1-only reporters were created by replacing the *SalI-BstEII* fragment of WT *GCN4-lacZ* construct p180[44] with the appropriate *SalI-BstEII* fragments, generated by DNA synthesis by LifeSct LLC, containing point mutations in the start codons of uORFs 2-4 (*uORF2* ATG to CTG, *uORF3* ATG to AGG, and *uORF4* ATG to AGG) and either WT uORF1 (plasmid pSG61) or uORF1 variants with altered third codons (pSG62-pSG89). The inserts in the resulting plasmids were sequenced in their entirety to confirm the desired sequences. The sequence of the *SalI-BstEII* fragment of pSG61 (with WT uORF1) is listed in Supplementary Data 3; sequences of all other constructs differ only by the replacements of the third uORF1 codon with the triplet indicated in Table S2 for the corresponding plasmids.

### *GCN4-lacZ* reporter assays
β-galactosidase specific activities (in units of nmol of ONPG cleaved per min per mg of protein) were assayed in whole cell extracts as described previously[33] in yeast transformants of uORF4-only and uSt-st *GCN4-lacZ* reporter plasmids, and as previously described[45] for transformants of uORF1-only reporter plasmids and the "uORF-less" *GCN4-lacZ* reporter p227[30]. Briefly, cells were transformed with plasmids bearing *GCN4-lacZ*

fusion reporters and grown in synthetic SD media to mid-log phase. Cells were centrifuged, washed with deionized water and broken using glass beads in Breaking Buffer (100 mM Tris 8.0, 4% Glycerol, 1 mM β-mercaptoethanol), then clarified by centrifugation (18,400 × *g*, 30 min, 4 °C). β-galactosidase activity (measured as nanomoles of o-nitrophenyl-3-Dgalactosidase cleaved per minute per milligram of protein) was normalized to protein concentration of the whole cell extract, which was measured by Bradford assay with BSA as standard.

For the uORF4-only and uSt-st-only reporter assays, the normalized β-galactosidase activity for each reporter construct in transformants of *tmaΔ* strains was determined by dividing the unnormalized β-galactosidase activities by a normalization factor. This factor was calculated separately for each individual experiment by assaying the activity of the "uORF-less" reporter (p227) in WT and *tmaΔ* strains in four transformants under the same conditions as uORF4-only/uSt-st-only reporters. Then, the mean of *tmaΔ* "uORF-less" activities was divided by the mean of WT "uORF-less" activities, determining the normalization factor.

For the uORF1-only reporter assays, β-galactosidase activities determined for each replicate transformant of the *tma20Δtma64Δ* strain were divided by a factor of 1.316 prior to calculating the mean activities to be compared to the mean unnormalized activities for the replicate transformants of the WT strain in a two-tailed, unpaired Student's *t* test to assign *p* values to differences between the means in mutant vs. WT cells. The normalization factor was determined by assaying six transformants each of the same *tma20Δtma64Δ* and WT strains harboring the "uORF-less" reporter (p227) under the same conditions employed for uORF1-only reporters. The *tma20Δtma64Δ*/WT ratio of mean expression values was 1.316, which differed significantly from unity in a two-tailed t-test with a p-value of 0.003.

### Yeast spotting assay
Yeast strains were spotted onto minimal synthetic defined (SD) medium plates in five serial 10-fold dilutions (starting with $OD_{600}$ 0.5) and grown for 48 h at 30 °C.

### Multiple sequence alignment and protein structure visualization
Multiple sequence alignment of *S. cerevisiae* Tma20 (UniProt accession P89886), *D. melanogaster* MCTS1 (UniProt accession Q9W445), *C. elegans* C11D2.7 (UniProt accession Q8MXH7) and *H. sapiens* MCTS1 (UniProt accession Q9ULC4) protein sequences was performed using Clustal Omega[46] with default settings through European Bioinformatics Institute Tools services[47]. Similarity scores and identity percentages for the sequences were determined using the UniProt BLAST tool.

Tma20 protein structure was predicted using the AlphaFold3 algorithm[48] through AlphaFold Server (Google DeepMind) services. MCTS1/DENR complex structure[39] was obtained through PDB accession 6MS4. PyMOL 3.0 was used for visualization of the structures.

### Statistics and reproducibility
Statistical significance was determined as follows. For all uORF4-only and Start-stop reporter assays, all data were tested for normality by the Shapiro–Wilk test. Based on the normality test, data were compared either by the parametric, unpaired, two-tailed Welch's *t* test, or by the nonparametric Mann–Whitney test; **** indicates $p < 0.0001$; *** $p < 0.001$; ** $p < 0.01$; * $p < 0.05$; ns non-significant. All individual data points are shown as part of bar plots and the number of replicates is shown above each bar. The SD was calculated from all biological replicates for each construct in each strain. For uORF1-only reporter assays, an unpaired, two-tailed Student's *t* test was employed. Statistical analysis and visualization were performed in GraphPad Prism, version 9.4.1 (GraphPad Software). All replicates shown in the reporter assays graphs were independent transformants—biological replicates.

### Reporting summary
Further information on research design is available in the Nature Portfolio Reporting Summary linked to this article.

## Data availability

All data from this study are available within this paper and its Supplementary Information. All plasmids and strains used in this study are available upon request from the authors. Source data for graphs presented in the main figures can be found at Supplementary Data 1.

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

## Acknowledgements
We are thankful to all members of our laboratories for fruitful discussions. This work was supported in part by a Grant of Excellence in Basic Research (EXPRO 2019) provided by the Czech Science Foundation (19-25821X), the Praemium Academiae grant provided by the Czech Academy of Sciences, CZ.02.01.01/00/22_008/0004575 RNA for therapy by ERDF and MEYS (all to L.S.V.), by GA UK project no. 339022 by Charles University Grant Agency (to K.J.), and by the Intramural Program of the National Institutes of Health (S.G. and A.G.H.).

## Author contributions
St.Gu., L.S.V. and A.G.H. conceived and designed the project. K.J. and Sw.Ga. carried out majority of experiments and performed the data analysis; they were assisted by K.P. at the onset of this study. K.J., Sw.Ga., L.S.V. and A.G.H. interpreted the results. K.J., L.S.V. and A.G.H. wrote the paper with input from Sw.Ga.

## Funding

## Competing interests
The authors declare no competing interests.
