## [Transparent Peer Review file · Communications Biology]

Differential effects of 40S ribosome recycling factors on reinitiation at regulatory uORFs in GCN4 mRNA are not dictated by their roles in bulk 40S recycling

Corresponding Author: Dr Alan Hinnebusch

Version 0:

Reviewer comments:

Reviewer #1

(Remarks to the Author)

The manuscript titled "Impacts of yeast Tma20/MCTS1, Tma22/DENR and Tma64/eIF2D on translation reinitiation and ribosome recycling" by Kristína Jendruchová, Swati Gaikwad, Kristýna Poncová Poncová, Stanislava Gunišová, Leoš Shivaya Valášek, and Alan G. Hinnebusch; focuses on the study of Tma proteins functions in reinitiation and ribosome recycling of uORFs. Tma proteins are conserved, yet conflicting evidence suggest they might serve diverging function in reinitiation and ribosome recycling of uORFs from yeast to human. To answer these key questions, the authors examined the impact of Tma proteins on reinitiation of regulatory uORFs directing translational control of GCN4. Utilizing a variety of uORFs-GCN4-lacZ constructs, mutating the uORFs penultimate codon, reinitiation is investigated by analysis of reporter expression levels in *tma20Δ*, *tma22Δ*, *tma64Δ*, *tma20Δtma22Δ*, *tma20Δtma64Δ* and *tma20Δtma22Δtma64Δ* strains. The manuscript presents a highly detailed analysis of Tma factors, including penultimate codon dependency as well as the impact of the reinitiation potential of the specific uORF itself. Together, the results reveal crucial conclusions on the diverging functions Tma factors play in yeast in comparison to the human DENR/MCTS1 proteins in reinitiation and ribosome recycling.

I only have minor comments:

* In the methods section, it would be beneficial to provide a more detailed explanation of the main method - reporter expression levels analysis, including the units used (for example: β -galactosidase activities were expressed in units of nmol of o-nitrophenyl- β -d-galactopyranoside hydrolyzed/min/mg of protein).

* In several figures, including figure 6 for example, the authors normalize to uORF less construct the GCN4-LacZ expression levels. Why not normalize to the wt(TGC) uORF1 construct? Wouldn't comparing the most similar systems be more informative?

* Similar to table 1, I would suggest to add tables summarizing the fold-changes in all the reporters utilized in the study, to summarize the expression differences in tma mutants vs. WT strains.

***To study uORF1, we employed the "uORF1-only" reporter lacking the AUGs of uORFs 2-4". There is a space missing in ofuORFs.

* "We first examined the relative contribution of the Tma20/Tma22 heterodimer and Tma64 to the efficiency of REI following uORF4 translation, which we regard as an exemplar of REI-non-permissive uORFs in budding yeas".

In the last word a t is missing.

Reviewer #2

(Remarks to the Author)

COMMSBIO-24-1344_Review

Summary

The manuscript investigates the mechanisms through which the Tma20/Tma22 heterodimer and Tma64 proteins, orthologs of the mammalian MCTS1/DENR and eIF2D (respectively), engage in the process of translation reinitiation of uORFs (REI) and ribosome recycling. These were examined in vivo in yeast, since inconsistencies of reports performed in yeast extracts were contradictory to data published in mammalian cells.

During translation termination, the recycling of small ribosomal subunits involves the release of deacylated tRNA from the penultimate codon and the dissociation of the empty 40S subunit from the mRNA. The authors examined the effect of

changing the penultimate codons on reinitiation, those that confer dependence, or independence on Tma20/Tma22.

The authors used yeast mutants of genes encoding for the heterodimer which Tma20/Tma22 were deleted, along with Tma64. These mutants served for generation of mutants in which the penultimate coding were exchanged. REI occurs when the recycling of a ribosome post-translation is incomplete, allowing the 40S subunit to remain bound to the mRNA and initiate the translation of a downstream ORF through the recruitment of the ternary complex (TC). While it was found that yeast Tma20/Tma22 inhibit REI, the human DENR facilitates REI following the translation of specific short upstream open reading frames (uORFs).

This study investigates whether the removal of Tma proteins influences reinitiation at specific uORFs within GCN4 mRNA. These transcripts include uORF1, which suppresses recycling and facilitates reinitiation, and uORF4, which enhances recycling and inhibits reinitiation. Wild-type cells have shown that specific penultimate codons lead to a significantly higher accumulation of unrecycled 40S subunits in mutant cells, a phenomenon referred to as Tma-dependence. In this study, the researchers generated original uORF1 and uORF4, as well as variants with penultimate codons that exhibit Tma-dependence in bulk 40S recycling. Their findings indicate that Tma proteins consistently decrease reinitiation at both the original uORF4 and its variants, irrespective of the presence of Tma-dependence in bulk 40S recycling. However, Tma proteins do not affect the reinitiation process at any form of uORF1.

This study highlights the varied impacts of Tma proteins on the reinitiation process, depending on the reinitiation potential of the uORF and the penultimate codon. Distinct from observations in mammals, these effects on reinitiation do not depend on the codons' Tma-dependency in bulk 40S recycling.

The study is structured on well-established biochemical principles and leverages a combination of genetic and molecular biology tools to dissect the contributions of these proteins to post-termination ribosomal processes. The topic is of considerable interest as it explores the evolutionary variances in the mechanism of translation reinitiation between yeast and higher eukaryotes. The authors present a comprehensive analysis, including the utilization of GCN4-lacZ reporters harboring mutations at various positions to dissect the roles of Tma factors in both promoting and inhibiting ribosome recycling and reinitiation. This study highlights the complexity of translational control and offers novel insights into the modulation of gene expression at the translational level.

While the research presents several strengths, including robust experimental design and clear logical progression through the questions addressed, certain aspects require additional clarification prior to publication.

The authors write that they examined the effects of eliminating the Tma proteins on reinitiation at the key regulatory uORFs mediating translational control of GCN4 mRNA that are optimized for impeding recycling and promoting reinitiation (uORF1) or for promoting recycling and impeding reinitiation (uORF4). They report that the Tma proteins reduce reinitiation at both uORF4 and uORF4 variants, regardless of whether the codons confer Tma dependence. They also report that the Tma factors had no effect on reinitiation on GCN4 mRNA, even if they were equipped with penultimate codons known to confer Tma dependence for reinitiation on GCN4 mRNA.

The authors also indicate that converting uORF1 or uORF4 to AUG-stop elements did not result in Tma dependence.

The final sentence of their abstract is that unlike in mammals, the effects on reinitiation are not dictated by the Tma-dependence of the codon in bulk 40S recycling.

Specific remarks to be addressed prior to publication.

1. Overall the paper is difficult to follow. It therefore requires substantial re-writing to advance clarity. The results occasionally varied from the authors' expectations, and the explanations to these variations were unclear.

2. Title

The title is descriptive and does not convey the conclusions that the authors wish to convey to the readers. It is in line with the difficulties to understand what these are.

3. Abstract

The abstract effectively introduces the study and clearly communicates the findings. However, there's an inconsistency in abbreviations. For instance, "upstream open reading frames" is abbreviated as "uORFs," but "reinitiation" lacks a corresponding abbreviation. It's advisable to review the entire text for similar inconsistencies to ensure uniformity.

4. Introduction

The Introductory chapter is overly lengthy and complex, includes information which is not fully relevant and therefore makes it difficult to focus on the research goals. Figure 1 is unclear. It attempts to summarize data that were previously published, but it also includes models that they would like to advance. Such models cannot be presented as part of the introduction and should be moved to the Discussion chapter. The authors should also indicate how their research advanced new models. This is not conveyed by the title, as indicated above. The Discussion also fails to deal with this in a clear manner.

5. Figures

Figure 1 - is not clear and hard to follow. It needs to be simplified for better understanding. The models should be presented after the authors reported their results. In such case, figure numbers should be adjusted.

Figure 2 – The authors discuss the option of leaky scanning, but do not describe such results with respect to uORF1 and/or uORF4.

Figure 3: In the figure legend it is written "each experiment was repeated at least once, and a representative experiment is presented here. The Figure should present the data of at least three independent repeats that served to generate the standard deviation which is shown in the figure. Currently, it is also not clear how the authors performed a statistical analysis for calculation of the SD from two experimental repeats that they mention.

Figure 3 shows that although a single mutation in the Tma genes had no effect on expression of the reporter protein, the triple mutation resulted in X3 increase in its expression. The authors provide an explanation that they also show in Figure 1, but it is highly speculative and does not justify to be included as part of their suggested model.

Regarding the statistical analysis, the authors should perform a statistical test in addition to the calculation of SD. Once done, the authors should indicate the significance of the differences that they follow, using asterisks to denote levels of significance or "ns" for not significant.

Figure 4 – The histograms should present the results obtained from 3 independent experiments.

Figure 7 – please add the statistical analysis for adding the significance of the results.

Figure 8: It would be beneficial to add similarity scores comparing *S. cerevisiae* to other orthologs. If feasible, employ AlphaFold for structural comparisons to potentially uncover functional differences between yeast and human proteins through domain or structural predictions.

Also – please change the color of the "red dot" indicating conservation between K and Q residues. Light red should do.
Supplementary Table S2: It would be beneficial to restructure Table S2, dividing it into sections based on the uORFs (1 or 4) and categorizing them by neutral, dependent, or independent characteristics for clearer description. The reference column could be eliminated, with references instead being incorporated into the description column.

6. Discussion

Should be shortened and much better focused. Special attention should be given to the final conclusions of this study which the authors fail to deliver. The proposed model could fit here in this chapter.

A second point that requires attention is how the single deletion of either Tma20 or Tma22 affects the expression of its counterpart protein in the heterodimer complex.

Page 19: it was difficult to follow the logic of the following text:

"Even though the median $tma20\Delta tma22\Delta tma64\Delta/WT$ derepression ratio is significantly larger for the reporters with Tma-dependent vs. Tma-independent codons (Fig. 5A), the Tma-dependence in bulk 40S recycling determined previously¹⁶ is generally a weak predictor of Tma dependence in promoting 40S recycling and suppressing REI for different penultimate codons at uORF4 (Table 1). Recycling and REI at the uORF4 variant containing the Tma-independent TGG Trp codon appeared to be the least dependent on Tma factors among all 13 different penultimate codons we examined. While the underlying reason remains unknown, it is interesting that TGG Trp is the sole codon for tryptophan."

This was one example. However, the Discussion is difficult to follow and to extract the major conclusions of this study, which appear to be important, yet much of the contributions delivered by the authors are lost among the multiple details. Too many dense trees mask the forest!

7. Minor points:

On page 9, there is a missing space in the word "ofuORFs."

Page 9 – "yeast" instead of "yeas" (row 2 from bottom)

There is a typo on page 26; "Nurseothricine" should be corrected to "Nourseothricine."

Reviewer #3

(Remarks to the Author)

The GCN4 mRNA in *S. cerevisiae* has proved to be an outstanding model for understanding mechanisms of translation initiation and reinitiation following translation of short upstream ORFs (uORFs). Here Jendruchova and colleagues have used well established analysis systems to probe the role of 3 proteins Tma20/22/64 in the recycling of 40S subunits at the GCN4 uORFs. The Tma proteins are implicated in promoting removal of the final tRNA and 40S ribosomes following translation termination. Prior work using ribosome footprinting of 40S at stop codons has identified Tma-dependent codons for main ORFs. Similar studies in mammalian cells have implicated Tma homologs in a subset of uORF controls.

The work here is generally well described, but as the authors are no doubt aware, is restricted to reporter gene analyses.

The take home message is that it is very unclear when the Tma factors are needed and any 'rules' taken from earlier studies don't seem to apply. Clearly a case where a lot more needs to be done to figure out what is happening. Nevertheless the data appears sound and should provoke discussion and further experimentation. I have the following points to raise...

1. There is no description of any attempt to compare the codon specificities identified with decoding rates, codon adaptation indices, codon-pair preferences etc. There are lots of studies of these parameters. Were any comparisons made?
2. The authors use 4 terms (mostly 3) to define the codon dependence. Tma-dependent is clear, but the terms Tma-neutral and Tma-independent could be thought interchangeable. I had to refer to the original Guydosh paper to figure out what was meant. Those termed Tma-independent here, were in that ribosome footprinting study actually sensitive to Tma presence. I.e. the 40S footprints were significantly lower in the Tma mutant than in the WT suggesting that the Tma factors impede 40S recycling on these templates. So potentially 'Tma-sensitive' rather than independent.

It would help readers if the text could be made clearer when these terms are introduced about what they actually refer to. The 4th term used is Tma 'NOT dependent' which appears in Figure 3 and is presumably the same as Tma-independent as stated in the Figure legend? Changing this to be consistent is recommended.
3. P10 line 4. The text states isogenic WT data are shown in Figure 3B but there is no data for WT uORF4 (ending with CCGPro) shown in these strains vs WT.
4. The start-stop ORF data is intriguing. Does this suggest a distinction between the first post-initiation step when initiator tRNA is in the P site vs later when elongation-specific Met-tRNA_{met} is used?
5. The methods describes that the beta-gal data were normalised to an orfless control, presumably to control for RNA stability effects? This is not obvious from the main text and adding sentences to this effect would be useful.
6. For each plot it would be better to show individual replicate values rather than mean \pm SD, or to supply the plotted data as an excel supplementary file.

Minor

1. Typo p9 'yeas'
2. Table 1 should report numbers \pm error and use '.' Rather than ',' for decimal points.
3. P14 5 lines up 'penultimate tRNA'. Surely the penultimate codon is decoded by the final tRNA?

Author Rebuttal letter:

We would like to thank the reviewers for their thoughtful comments and suggestions. We have tried to address all of them with the appropriate additional analyses or revisions of figures or text, as described fully below.

Reviewer #1 (Remarks to the Author):

I only have minor comments:

* In the methods section, it would be beneficial to provide a more detailed explanation of the main method - reporter expression levels analysis, including the units used (for example: β -galactosidase activities were expressed in units of nmol of o-nitrophenyl- β -d-galactopyranoside hydrolyzed/min/mg of protein).
> We expanded our explanation of the method.

* In several figures, including figure 6 for example, the authors normalize to uORF less construct the GCN4-LacZ expression levels. Why not normalize to the wt(TGC) uORF1 construct? Wouldn't comparing the most similar systems be more informative?

> Normalization to the uORF-less construct was chosen to control for any changes in general translation that deletion of the Tma genes could cause. In this way, we could directly monitor the changes in recycling vs. reinitiation rates caused by the individual deletions.

* Similar to table 1, I would suggest to add tables summarizing the fold-changes in all the reporters utilized in the study, to summarize the expression differences in tma mutants vs. WT strains.

> We added uORF1 reporter constructs and the corresponding fold-changes of their reporter expression in tma20 Δ tma64 Δ mutant versus WT to Table 1.

** "To study uORF1, we employed the "uORF1-only" reporter lacking the AUGs of uORFs 2-4". There is a space missing in ofuORFs.

> Corrected.

* "We first examined the relative contribution of the Tma20/Tma22 heterodimer and

Tma64 to the efficiency of REI following uORF4 translation, which we regard as an exemplar of REI-non-permissive uORFs in budding yeas”.

In the last word a t is missing.

> Corrected.

Reviewer #2 (Remarks to the Author):

1. Overall the paper is difficult to follow. It therefore requires substantial re-writing to advance clarity. The results occasionally varied from the authors' expectations, and the explanations to these variations were unclear.

>As described below, we have divided Fig. 1 into two figures such that Fig. 1 now only describes the predicted outcomes of our experiments based on published results on the effects of MCTS1/DENR on REI and prior work on typical uORFs in yeast. The new Fig. 8 describes our models to account for the discrepancies between the predicted and observed outcomes, which are presented in the Discussion. We have also edited both Introduction and Discussion to make it easier to grasp the predicted outcomes, our findings that depart from those predictions, and the novel functions we ascribe to the Tma factors in regulating REI at GCN4 uORF1 and uORF4.

2. Title

The title is descriptive and does not convey the conclusions that the authors wish to convey to the readers. It is in line with the difficulties to understand what these are.

>As we stated clearly in the abstract, our study has two main conclusions: “Thus, effects of the Tma proteins vary depending on the REI potential of the uORF and the penultimate codon, but unlike in mammals, are not principally dictated by the Tma-dependence of the codon in bulk 40S recycling.” We agree with the referee and have revised the title to capture better the main findings of the study as: “Differential effects of 40S ribosome recycling factors on reinitiation at regulatory uORFs in GCN4 mRNA are not dictated by defects in bulk 40S recycling”.

3. Abstract

The abstract effectively introduces the study and clearly communicates the findings. However, there's an inconsistency in abbreviations. For instance, "upstream open reading frames" is abbreviated as "uORFs," but "reinitiation" lacks a corresponding abbreviation. It's advisable to review the entire text for similar inconsistencies to ensure uniformity.

>The abbreviation for reinitiation (REI) was defined in the Abstract and used throughout.

4. Introduction

The Introductory chapter is overly lengthy and complex, includes information which is not fully relevant and therefore makes it difficult to focus on the research goals. Figure 1 is unclear. It attempts to summarize data that were previously published, but it also includes models that they would like to advance. Such models cannot be presented as part of the introduction and should be moved to the Discussion chapter. The authors should also indicate how their research advanced new models. This is not conveyed by the title, as indicated above. The Discussion also fails to deal with this in a clear manner.

>We simplified Fig. 1 to restrict it to summarizing the published results on the consequences of depleting DENR on REI at short uORFs containing DENR-dependent or -independent codons, and the expectations for typical yeast uORFs. The revised Fig. 1 is sufficient to facilitate understanding of the published data described in the Introduction and the attendant predictions of the consequences of introducing different penultimate codons at yeast uORF1 and uORF4 that are examined experimentally in RESULTS. A new figure (revised Fig. 8) is being presented to facilitate our description of proposed models to explain departures of the observed data from expected results in RESULTS and DISCUSSION. This modification of figures should make it easier for the reader to grasp the predicted outcomes of our experiments based on the previous findings on mammalian DENR/MCTS1 (Fig. 1) as well as the revised models we propose to explain the discrepancies between those predictions and our observed data (Rev. Fig. 8). The final paragraph of the Introduction clearly summarizes our main findings, indicating how they depart from the expectations-based studies of DENR/MCTS1 and the additional functions we ascribe the yeast Tma factors in regulating reinitiation. As noted above, we have modified the manuscript title to capture these novel insights.

5. Figures

Figure 1 - is not clear and hard to follow. It needs to be simplified for better understanding. The models should be presented after the authors reported their results. In such case, figure numbers should be adjusted.

>Done—see our responses to the previous point.

Figure 2 – The authors discuss the option of leaky scanning, but do not describe such results with respect to uORF1 and/or uORF4.

> On page 9, we had cited previous experimentation establishing that leaky scanning is extremely infrequent at both native uORF1 and native solitary uORF4, but we have now modified this passage to make it clearer.

Figure 3: In the figure legend it is written "each experiment was repeated at least once, and a representative experiment is presented here. The Figure should present the data of at least three independent repeats that served to generate the standard deviation which is shown in the figure. Currently, it is also not clear how the authors performed a statistical analysis for calculation of the SD from two experimental repeats that they mention.

> Firstly, we want to apologize for an unclear statement in the figure legend. In fact, all experiments presented in this study were performed with at least 3 biological replicates. In other words, each yeast colony that we picked for analysis comes from a separate transformation and cultivation in a flask. We have no technical replicates. All experiments were carried out with at least n=3 biological replicates each; the number of bio-replicates is now clearly specified in each figure legend. For this particular figure, all experiments were carried out with at least n=5 biological replicates each.

We have improved our statistical analysis of the results obtained for all biological replicates of the different uORF4-only reporters, which we now express as units of β -galactosidase activity in WT and deletion strains, instead of % relative to WT strain. All reporter data were first tested for normality of distribution by the Shapiro-Wilk test. Based on the normality test, differences between expression in WT versus tma deletions strains (or between two different tma deletion strains, where applicable) were compared either by the parametric, unpaired, two-tailed Welch's t test (for those that passed the normality test), or by the non-parametric Mann-Whitney test (for those that did not). All individual data points are now shown as part of bar plots. The SD was calculated from all biological replicates for each construct in each strain. We modified the RESULTS and METHODS accordingly.

Figure 3 shows that although a single mutation in the Tma genes had no effect on expression of the reporter protein, the triple mutation resulted in X3 increase in its expression. The authors provide an explanation that they also show in Figure 1, but it is highly speculative and does not justify to be included as part of their suggested model.

> With all due respect to this reviewer, this is a misunderstanding. As we reported, the single deletion of TMA64 showed a much lower and less significant effect on reporter expression; single deletions of TMA20 and TMA22 lead to ~2x increased expression of the reporters. These findings are consistent with the ribosome profiling data of Guydosh et al. 2021, where a single deletion of TMA20 or TMA22 resulted in a similar codon dependency as the tma20 Δ /tma64 Δ double deletion, which was not the case for the single TMA64 deletion.

Regarding the statistical analysis, the authors should perform a statistical test in addition to the calculation of SD. Once done, the authors should indicate the significance of the differences that they follow, using asterisks to denote levels of significance or "ns" for not significant.

> This point is addressed above. We denoted the statistical significance in the plots with asterisks in all figures; **** indicates $p < 0.0001$; *** $p < 0.001$; ** $p < 0.01$; * $p < 0.05$; ns = non-significant.

Figure 4 – The histograms should present the results obtained from 3 independent experiments.

As explained above, all our repetitions represent individual biological replicates. For this particular figure, all experiments were carried out with at least n=4 biological replicates each, which are all shown in histograms as individual data points. Additionally, for better clarity of our data, we now reordered the results and divided the constructs into new panels B, C and D

according to their Tma-dependence; newly defined as Tma-hypodependent, Tma-average and Tma-hyperdependent (as explained to Reviewer #3 below). We modified the RESULTS and METHODS accordingly. Based on this reanalysis, we also made slight corrections to calculations of data in Table 1 and Figure 5.

Figure 7 – please add the statistical analysis for adding the significance of the results. We now added statistical analysis for our reporter measurements in Fig. 7, performed the same way as for Figures 3 and 4. We also edited the graphs to unify the approach with Figures 3, 4 and 6. For better clarity, we plotted values for uORF1 in separate plots, since the reporter enzyme activity of uORF1 reaches much higher values than of other constructs.

Figure 8: It would be beneficial to add similarity scores comparing *S. cerevisiae* to other orthologs. If feasible, employ AlphaFold for structural comparisons to potentially uncover functional differences between yeast and human proteins through domain or structural predictions. Also – please change the color of the "red dot" indicating conservation between K and Q residues. Light red should do.

> We employed the newly released AlphaFold 3 (Abramson et al. 2024) to predict the structure of yeast Tma20p. Comparison of the discussed conserved site in the predicted Tma20p structure with published human MCTS1 structure can be seen in new Fig. 9 (formerly Fig. 8), along with similarity scores and Identity (%) of yeast Tma20 with its other orthologs.

Supplementary Table S2: It would be beneficial to restructure Table S2, dividing it into sections based on the uORFs (1 or 4) and categorizing them by neutral, dependent, or independent characteristics for clearer description. The reference column could be eliminated, with references instead being incorporated into the description column.

> We restructured Table S2 according to the reviewer's suggestions.

6. Discussion

Should be shortened and much better focused. Special attention should be given to the final conclusions of this study which the authors fail to deliver. The proposed model could fit here in this chapter.

>We have removed unnecessary details and statements to streamline the DISCUSSION and, as described above, added the new Figure 8 to help explain the models presented there to fully explain the roles of Tma factors in regulating REI at GCN4 uORF1 and uORF4. The final paragraph of the Discussion now succinctly captures our key findings and their implications for distinct roles of the Tma factors in regulating REI in yeast, which differ from the those ascribed to mammalian MCTS1/DENR.

A second point that requires attention is how the single deletion of either Tma20 or Tma22 affects the expression of its counterpart protein in the heterodimer complex.

> This was established for mammalian DENR/MCTS1, showing that knockout of DENR (Tma22) dramatically lowers protein levels of MCTS1 (Tma20) and vice versa (Bohlen et al. 2020, Fig. 2A), and we assume that it is the same for their yeast homologues. In any case, with all due respect to this reviewer, we do not think that this particular information is critical for our conclusions because: 1) in Fig. 3, where both Tma20 and Tma22 single deletions were tested, they both behaved the same, and 2) for all other experiments we used either the tma20/22 double deletion or the tma20/64 deletion, the latter of which behaves the same as the tma triple deletion, as we described even in the original manuscript (Fig. 3C).

Page 19: it was difficult to follow the logic of the following text:

"Even though the median $tma20\Delta tma22\Delta tma64\Delta/WT$ derepression ratio is significantly larger for the reporters with Tma-dependent vs. Tma-independent codons (Fig. 5A), the Tma-dependence in bulk 40S recycling determined previously¹⁶ is generally a weak predictor of Tma dependence in promoting 40S recycling and suppressing REI for different penultimate codons at uORF4 (Table 1). Recycling and REI at the uORF4 variant containing the Tma-independent TGG Trp codon appeared to be the least dependent on Tma factors among all 13 different penultimate codons we examined. While the underlying reason remains unknown, it is interesting that TGG Trp is the sole codon for tryptophan."

This was one example. However, the Discussion is difficult to follow and to extract the major conclusions of this study, which appear to be important, yet much of the contributions delivered by the authors are lost among the multiple details. Too many dense trees mask the forest!

>We re-worded the first sentence cited here to make it more transparent and removed the second and third in our efforts to streamline the DISCUSSION.

7. Minor points:

On page 9, there is a missing space in the word "ofuORFs."

> Corrected.

Page 9 – "yeast" instead of "yeas" (row 2 from bottom)

> Corrected.

There is a typo on page 26; "Nurseothricine" should be corrected to "Nourseothricine."

> Corrected.

Reviewer #3 (Remarks to the Author):

I have the following points to raise...

1. There is no description of any attempt to compare the codon specificities identified with decoding rates, codon adaptation indices, codon-pair preferences etc. There are lots of studies of these parameters. Were any comparisons made?

> As advised, we compared the decoding rates (expressed as Ribosome Resting Rates (RRT) as per Gardin et al. 2014) and codon frequencies per 1000 codons (taken from the CoCoPUTs database) of our Tma-hyper, hypo or average dependent codons. We were unable to compare codon adaptation indices, since these are calculated for individual genes, not codons. We found that Tma-average codons have significantly higher RRT than Tma-hyperdependent codons and higher, albeit not significantly, RRT than Tma-hypodendent codons, as shown below in Figure S1 presented only for the reviewers/editor. We also tested the differences between codon frequencies and found that Tma-hyperdependent codons have significantly higher codon frequencies than both Tma-hypodendent and Tma-average codons, as also shown below. We also found (predictably) a negative correlation between RRT and codon frequencies (below). However, we could not think of any logical connection between these analyses and our results. Therefore, we did not include it in the revised manuscript. The most probable reason for that is that all these parameters reflect the situation on the elongating ribosome with a coding codon occupying its A-site. Our data, however, apply to terminating ribosomes with a stop codon in the A-site. It is needless to mention that these molecular species are qualitatively very different.

2. The authors use 4 terms (mostly 3) to define the codon dependence. Tma-dependent is clear, but the terms Tma-neutral and Tma-independent could be thought interchangeable. I had to refer to the original Guydosh paper to figure out what was meant. Those termed Tma-independent here, were in that ribosome footprinting study actually sensitive to Tma presence. I.e. the 40S footprints were significantly lower in the Tma mutant than in the WT suggesting that the Tma factors impede 40S recycling on these templates. So potentially 'Tma-sensitive' rather than independent.

It would help readers if the text could be made clearer when these terms are introduced about what they actually refer to. The 4th term used is Tma 'NOT dependent' which appears in Figure 3 and is presumably the same as Tma-independent as stated in the Figure legend? Changing this to be consistent is recommended.

> The reviewer makes an excellent point. In response, we have added text to RESULTS indicating how Tma-dependence was determined in the published experiments of Ref. 16 and also changed our nomenclature from "Tma-dependent, Tma-independent, and Tma-neutral" to "Tma-hyperdependent, Tma-hypodependent and Tma-average", respectively. These labels are more accurate because even the least Tma-dependent codons show measurable defects in bulk 40S recycling in the translatome that are merely smaller than average in degree¹⁶: "Tma-dependence for each triplet was assigned previously as the ratio of 40S occupancies in tma20Δtma64Δ vs. WT cells averaged across the stop codons of all genes¹⁶. Here, we designate penultimate codons as being hyper- or hypodependent on Tma factors for recycling if they showed, respectively, greater or less than average tmaΔΔ/WT 40S occupancy ratios at a 99% confidence level, with all other triplets

exhibiting average Tma-dependency.” We have also modified the text and figures throughout to refer more accurately to different degrees of Tma-dependency rather than describing Tma-dependence, Tma-independence, or Tma-neutrality.

3. P10 line 4. The text states isogenic WT data are shown in Figure 3B but there is no data for WT uORF4 (ending with CCGPro) shown in these strains vs WT.

> This is just a misunderstanding. ‘Isogenic’ in this context means that all strains tested have the same genetic background; it does not concern the reporters.

4. The start-stop ORF data is intriguing. Does this suggest a distinction between the first post-initiation step when initiator tRNA is in the P site vs later when elongation-specific Met-tRNA^{Met} is used?

> We believe that translation termination that immediately follows the initiation step is mechanistically different from that following the full elongation cycle. In fact, we are extensively investigating the differences between these two processes in budding yeast and human cell lines. It is likely that this difference is reflected in the present data, but it has yet to be firmly established.

5. The methods describes that the beta-gal data were normalised to an orfless control, presumably to control for RNA stability effects? This is not obvious from the main text and adding sentences to this effect would be useful.

> As explained to Rev. #1 above, normalization to the uORF-less construct was chosen to control for any changes in general translation that might be conferred by deletion of the TMA genes. In this way, we could directly monitor the changes in recycling vs. reinitiation rates caused by the individual deletions.

6. For each plot it would be better to show individual replicate values rather than mean \pm SD, or to supply the plotted data as an excel supplementary file.

> As explained to Reviewer #2 above, we have greatly improved our statistical analysis of biological replicates for all uORF4-only reporter assays, which we expressed as units of β -galactosidase activity in WT and deletion strains, instead of % relative to WT strain. All reporter data were first tested for normality of distribution by the Shapiro-Wilk test. Based on the normality test, differences between expression in WT versus tma deletions strains (or between two different tma deletion strains, where applicable) were compared either by the parametric, unpaired, two-tailed Welch’s t test (for those that passed the normality test), or by the non-parametric Mann-Whitney test (for those that did not). All individual data points are now shown as part of bar plots. The SD was calculated from all biological replicates for each construct in each strain. We modified the RESULTS and METHODS accordingly.

Minor

1. Typo p9 ‘yeas’

> Corrected.

2. Table 1 should report numbers \pm error and use ‘.’ Rather than ‘,’ for decimal points.

> We modified the Table according to the reviewer’s suggestions.

3. P14 5 lines up ‘penultimate tRNA’. Surely the penultimate codon is decoded by the final tRNA?

> We corrected this error and refer instead to the tRNA decoding the penultimate codon, abbreviated in Figs. 1 & 8 legends as the “penultimate-codon tRNA”.

Version 1:

Reviewer comments:

Reviewer #1

(Remarks to the Author)

In their revision of the manuscript titled “Differential effects of 40S ribosome recycling factors on reinitiation at regulatory uORFs in GCN4 mRNA are not dictated by their roles in bulk 40S recycling” by Kristína Jendruchová, Swati Gaikwad, Kristýna Poncová Poncová, Stanislava Gunišová, Leoš Shivaya Valášek, and Alan G. Hinnebusch;

The authors have addressed all my comments and suggestions. The manuscript represents a novel, significant, and insightful contribution that advances the field. It will be of great interest to the readers of Communications Biology and can be

published as is.

Reviewer #2

(Remarks to the Author)

The authors have answered all the points I raised in my original review. The title was indeed changed, although it is still quite descriptive. I assume that this is due to the nature of the results, which made it difficult to reach a conclusive statement. However, since the topic is important, I will not recommend to change it.

Reviewer #3

(Remarks to the Author)

The authors have submitted a considerably revised version of their manuscript with changes to both text and figures. In my view these changes have suitably addressed the major points raised by the reviewers. The revised nomenclature for Tma-dependence and its explanation as well as the revised model figures should all enhance the readability of the manuscript to a wider audience.

I have no further concerns.
